# Neural Basis Models for Interpretability

**Filip Radenovic**
Meta AI

**Abhimanyu Dubey**
Meta AI

**Dhruv Mahajan**
Meta AI

## Abstract

Due to the widespread use of complex machine learning models in real-world applications, it is becoming critical to explain model predictions. However, these models are typically black-box deep neural networks, explained post-hoc via methods with known faithfulness limitations. Generalized Additive Models (GAMs) are an inherently interpretable class of models that address this limitation by learning a non-linear shape function for each feature separately, followed by a linear model on top. However, these models are typically difficult to train, require numerous parameters, and are difficult to scale. We propose an entirely new subfamily of GAMs that utilizes basis decomposition of shape functions. A small number of basis functions are shared among all features, and are learned jointly for a given task, thus making our model scale much better to large-scale data with high-dimensional features, especially when features are sparse. We propose an architecture denoted as the Neural Basis Model (NBM) which uses a single neural network to learn these bases. On a variety of tabular and image datasets, we demonstrate that for interpretable machine learning, NBMs are the state-of-the-art in accuracy, model size, and, throughput and can easily model all higher-order feature interactions. Source code is available at `github.com/facebookresearch/nbm-spam`.

## 1 Introduction

Real world machine learning models [14, 61] are mostly used as a *black-box*, *i.e.*, it is very difficult to analyze and understand why a specific prediction was made. In order to *explain* such black-box models, an instance-specific local interpretable model is often learned [38, 50]. However, these approaches tend to be unstable and unfaithful [1, 52], *i.e.*, they often misrepresent the model's behavior. On the other hand, a family of models known as generalized additive models (GAMs) [24] have been used for decades as an inherently interpretable alternative to black-box models.

GAMs learn a *shape function* for each feature independently, and outputs of such functions are added (with a bias term) to obtain the final model prediction. All models from this family share an important trait: the impact of any specific feature on the prediction does not rely on the other features, and can be understood by visualizing its corresponding shape function. Original GAMs [24] were fitted using splines, which have since been improved in explainable boosting machines (EBMs) [36] by fitting boosted decision trees, or very recently in neural additive models (NAMs) [2] by fitting deep neural networks (DNNs). A drawback for all the aforementioned approaches is that for each shape function, they require either millions of decision trees [36], or a DNN with tens of thousands of parameters [2], making them prohibitively expensive for learning datasets with a large number of features.

In this work, we propose a novel subfamily of GAMs, which, unlike previous approaches, learn to decompose each feature's shape function into a small set of basis functions *shared* across all features. The shape functions are fitted as the feature-specific linear combination of these shared bases, see Figure 1. At an abstract level, our approach is motivated by signal decomposition using traditional basis functions like the Fourier basis [9] or Legendre polynomials [45], where a weighted

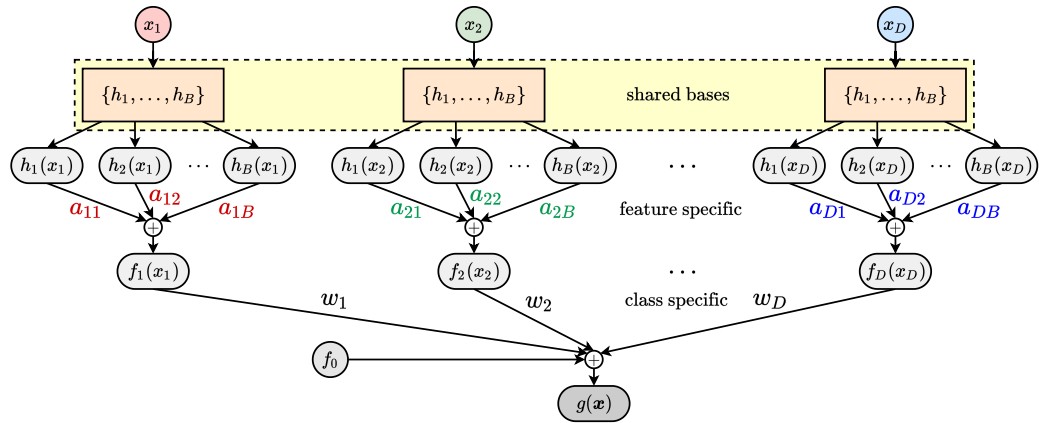

Figure 1: Neural Basis Model (NBM) architecture for a binary classification task.

combination of a few basis functions suffice to reconstruct complex signal shapes. However, in contrast to these approaches, our basis decomposition is not fixed *a priori*. In fact, it is learnt specifically for the prediction task. Consequently, we maintain the most important feature of GAMs, *i.e.*, their interpretability, as the contribution of single feature does not depend on the other features. At the same time, we gain scalability, as the number of basis functions needed in practice is much smaller than the number of input features. Moreover, we show that the usage of basis functions can increase computational efficiency by several orders of magnitude when the input features are sparse. Additionally, we propose an approach to learning the basis functions using a single DNN. We call this solution the Neural Basis Model (NBM). Using neural networks allows for even higher scalability, as training and inference are performed on GPUs or other specialized hardware, and can be easily implemented in any deep learning framework using standard, already developed, building blocks.

Our contributions are as follows: (i) We propose a novel subfamily of GAMs whose shape functions are composed of shared basis functions, and propose an approach to learn basis functions via DNNs, denoted as Neural Basis Model (NBM). This architecture is suitable for mini-batch gradient descent training on GPUs and easy to plug-in into any deep learning framework. (ii) We demonstrate that NBMs can be easily extended to incorporate pairwise functions, similar to GA$^2$Ms [37], by learning another set of bases to model the higher order interactions. This approach effectively only linearly increases the parameters, while other models such as EB$^2$Ms [36, 37] and NA$^2$Ms [2] suffer from quadratic growth of parameters, and often require heuristics and repeated training to select the most important interactions before learning [37]. (iii) Through extensive evaluation of regression, binary classification, and multi-class classification, with both tabular and computer vision datasets, we show that NBMs and NB$^2$Ms outperform state-of-the-art GAMs and GA$^2$Ms, while scaling significantly better, *i.e.*, fitting much fewer parameters and having higher throughput. For datasets with more than ten features, using NBMs result in around $5\times$–$50\times$ reduction in parameters over NAMs [2], and $4\times$–$7\times$ better throughput. (iv) We propose an efficient extension of NBMs to sparse datasets with more than a hundred thousand features, where other GAMs do not scale at all.

## 2 Related work

Shape functions in GAMs [24, 64] have many different representations in the literature, including: splines [24], trees or random forests [36], deep neural networks [2], neural oblivious decision trees [12]. Popular methods of fitting GAMs are: backfitting [24], gradient boosting [36], or mini-batch gradient descent [2, 12]. Our work falls under the GAM umbrella, however, it differs from these approaches by not learning shape functions independently, but rather, learning a set of shared basis functions that are used to compose each shape function. The bases themselves can be complex non-linear functions that operate on one feature at a time, *e.g.*, decision trees or splines, however, to keep the scope of the work concise and to ensure straightforward scalability, we represent them with deep neural networks and use mini-batch stochastic gradient descent to learn the bases. This makes our work most closely related to neural additive models (NAMs) [2], however, crucially, our method learns far fewer parameters by sharing bases compared to NAMs.

Methods have been proposed to model pairwise interactions [12, 37, 58], and they are commonly denoted as GA$^2$Ms. However, they usually require sophisticated feature selection using: heuristics [37], back-propagation [12], or, two-stage iterative training [58]. We note that one of the main goals of our work is to analyze the scalability of the newly proposed NBM architecture in extreme scenarios, hence, we do not apply feature selection algorithms. That being said, before-mentioned algorithms are complementary and can be applied to NB$^2$Ms as well.

Finally, GAMs are a popular choice in high-risk applications for healthcare [11, 56], finance [6], forensics [55], *etc.* Another line of work applicable to these domains are interpretable surrogate models such as LIME [50], SHAP [38], tree-based surrogates [5, 57], that give a *post-hoc* explanation of a high-complexity model. However, there are almost no theoretical guarantees that the simple surrogate model is highly representative of the more complex model [1]. This issue is completely resolved when using inherently transparent models such as NBMs. We hope that our approach sparks wider usage of GAMs in mission-critical applications with large-scale data, where the ability to interpret, understand, and correct the model is of utmost importance.

## 3 Method

### 3.1 Background

**Generalized Additive Model (GAM) [24].**   Given a $D$-dimensional interpretable input $\boldsymbol{x}=\{x_i\}_{i=1}^{D}$, $\boldsymbol{x} \in \mathbb{R}^D$, a target label $y$, a link function $g$ (*e.g.*, logistic function), a univariate shape function $f_i$, corresponding to the input feature $x_i$, a bivariate shape function $f_{ij}$, corresponding to the feature interaction, and a bias term $f_0$, the prediction function in GAM and GA$^2$M is expressed as

$$\textbf{GAM} : g(\boldsymbol{x}) = f_0 + \sum_{i=1}^{D} f_i(x_i); \quad \textbf{GA}^2\textbf{M} : g(\boldsymbol{x}) = f_0 + \sum_{i=1}^{D} f_i(x_i) + \sum_{i=1}^{D} \sum_{j>i}^{D} f_{ij}(x_i, x_j). \quad (1)$$

Interpreting GAMs is straightforward as the impact of a feature on the prediction does not rely on the other features, and can be understood by visualizing its corresponding shape function, *e.g.*, plotting $x_i$ on the $x$-axis and $f_i(x_i)$ on the $y$-axis. A certain level of interpretability is sacrificed for accuracy when modeling interactions, as $f_{ij}$ shape functions are harder to visualize. Shape function visualization through heatmaps [12, 37] is commonly used towards that purpose. Note that, the graphs visualizations of GAMs are an exact description of how GAMs compute a prediction.

### 3.2 Our model architecture

We observe that, typically, input features of high-dimensional data are correlated with each other. As a result, it should be possible to decompose each shape function $f_i$ into a small set of basis functions shared among all the features. This is the core idea behind our approach.

**Neural Basis Model (NBM).**   We propose to represent shape functions $f_i$ as

$$f_i(x_i) = \sum_{k=1}^{B} h_k(x_i)a_{ik}; \quad (2)$$

where $\{h_1, h_2, ..., h_B\}$ represents a set of $B$ shared basis functions that are independent of feature indices, and coefficients $a_{ik}$ are the projection of each feature to the shared bases. We additionally propose to learn basis functions using a DNN, *i.e.*, a single *one*-input $B$-output multi-layer perceptron (MLP) for all $\{h_k; k = 1, \dots, B\}$. The resulting architecture is shown in Figure 1.

**Multi-class / multi-task architecture.**   Let $l$ correspond to the target class $y_l$ in the multi-class setting. Similar to Equation 1, the prediction function $g_l$ for class $y_l$ in GAMs can be written as:

$$g_l(\boldsymbol{x}) = f_{0l} + \sum_{i=1}^{D} f_i(x_i)w_{il}, \quad (3)$$

where feature shape functions $f_i(x_i)$ are shared among the classes and are linearly combined using class specific weights $w_{il}$. Combining Equations 2 and 3, multi-class NBM can be represented as:

$$\textbf{Multi-class NBM} : g_l(\boldsymbol{x}) = f_{0l} + \sum_{i=1}^{D} \sum_{k=1}^{B} h_k(x_i)a_{ik}w_{il}. \quad (4)$$

**Extension to NB$^2$M.** Similar to NBM, we represent GA$^2$M shape functions $f_{ij}$ in Equation 1 as:

$$f_{ij}(x_i, x_j) = \sum_{k=1}^{B} u_k(x_i, x_j) b_{ijk};$$ (5)

where $\{u_1, u_2, ..., u_B\}$ represents a set of $B$ shared bi-variate basis functions that are independent of feature indices and coefficients $b_{ijk}$ are the projection of pair-wise features to the shared bases. We learn an additional *two*-input $B$-output MLP for all $\{u_k; k = 1, \dots, B\}$ to learn the bases. Extension to multi-class setting can be done in the same way as for NBMs.

**Sparse architecture.** Typically, datasets with high-dimensional features are sparse in nature. For example, in the Newsgroups dataset [32], news articles are represented by *tf-idf* features, and, for a given instance, most of the features are absent due to the vocabulary being of the order of 100K words. Since NBM uses a single DNN to learn all the bases, we can simply append the single value representing the absent feature to the batch, to compute the corresponding basis function values. The subsequent linear projection to feature indices via $a_{ik}$ is a computationally inexpensive operation.

In contrast, typical GAMs (*e.g.*, Neural Additive Model (NAM) [42]) need to pass the absent value through every shape function $f_i$ which makes it compute-intensive as well as difficult to implement.

**Training and regularization.** We use mean squared error (MSE) for regression, and cross-entropy loss for classification. To avoid overfitting, we use the following regularization techniques: (i) $L_2$-normalization (weight decay) [31] of parameters; (ii) batch-norm [28] and dropout [54] on hidden layers of the basis functions network; (iii) an $L_2$-normalization penalty on the outputs $f_i$ to incentivize fewer strong feature contributions, as done in [2]; (iv) basis dropout to randomly drop individual basis functions in order to decorrelate them. Similar techniques have been used for other GAMs [2, 12].

**Selecting the number of bases.** One can use the theory of Reproducing Hilbert Kernel Spaces (RKHS, [7]) to devise a heuristic for selecting the number of bases $B$. Specifically, we demonstrate that any NBM model lies on a subspace within the space spanned by a complete GAM if the GAM shape functions reside within a ball in an RKHS. Assuming a regularity property in the data distribution, one can then demonstrate that $B = \mathcal{O}(\log D)$ bases are sufficient to obtain competitive performance. We present this formally in Appendix Sec. A.5. This provides the alternate interpretation of NBM as learning a "principal components" decomposition in the $L_2-$space of functions, as we learn a set of (preferably orthogonal) basis functions to approximate the decision boundary.

### 3.3 Discussion

In this section, we contrast NBMs with closely related GAMs: Neural Additive Models (NAMs) [2].

**Neural Additive Model (NAM) [2].** NAMs learn a linear combination of networks that each attend to a single input feature: each $f_i$ in (1) is parametrized by a deep neural network, *i.e.*, a *one*-input *one*-output multi-layer perceptron (MLP). These MLPs are trained jointly using backpropagation and can learn arbitrarily complex shape functions.

**Number of parameters.** We compare number of weight parameters needed to learn NAM *vs.* NBM for the binary-classification task. See Appendix Section A.3 for multi-class analysis. Let us denote with $M$ the number of parameters in MLP for each feature in NAM, and with $N$ the number of parameters in MLP for bases in NBM. In most experiments the optimal NAM has 3 hidden layers with 64, 64 and 32 units ($M = 6401$), and, NBM has 3 hidden layers with 256, 128, 128 units ($N = 62820$) and $B = 100$ basis functions. Then the ratio of number of parameters in NAM *vs.* NBM is given by,

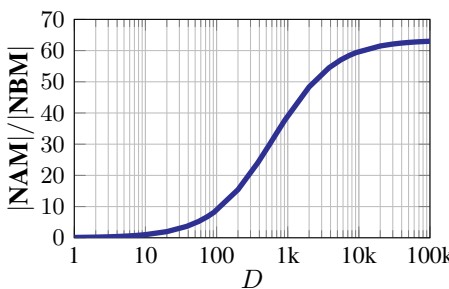

Figure 2: NAM *vs.* NBM: #parameters.

$$\frac{|\mathbf{NAM}|}{|\mathbf{NBM}|} = \frac{D \cdot M + D}{N + D \cdot B + D} = \frac{6402}{\frac{62820}{D} + 101}.$$ (6)

Figure 2 shows this ratio for different values of feature dimensionality $D$. For $D = 10$, NBMs and NAMs have roughly equal number of parameters. For most textual and vision datasets, feature dimensionality is significantly higher, thus leading to $10\times$–$50\times$ reduction in parameters. We also observe that as a result, for many datasets, specialized regularization discussed in the previous section gives incremental gains for NBMs, while they are very crucial for NAMs to give good performance. Additional analysis on number of parameters is given in Section 4.4 and Table 2.

**Throughput.**    One of the main challenges in approach like NAMs is the low throughput rate, that is the number of data instances processed per second, which directly affects the training speed. Since NAMs use separate MLP per dimension, it is much more challenging to implement efficiently. NBMs on the other hand are much more efficient since feature specific linear layer on the top of bases is very fast. For example, for datasets with around hundred features, the original NAM implementation [2] is around $20\times$ slower in training compared to NBMs. We optimized the speed of NAMs using group convolutions (see Appendix Section A.3), but even this optimized version is around $5\times$ slower.

**Stability.**    The interpretability of models and their explanations is tightly coupled to their stability. For example, post-hoc explanations of black-box models are known to be unstable, and produce different explanations for small input changes [53]. GAMs are exactly explained by visualizing shape functions that represent the model, however, a desirable property is to have similar shape functions when repeatedly training the model on the same data while varying random initialization. Because of the over-parameterization in NAMs, we observe that shape functions of different runs are often unstable (*i.e.* they often diverge), especially for feature regions where the data density is low. On the other hand, NBMs train a single set of shared bases for the entire dataset, which makes shape functions significantly more stable, see Section 4.6 and Figure 3.

**$NA^2M$ vs. $NB^2M$.**    NAMs can trivially be extended to $NA^2Ms$ by learning additional *two*-input *one*-output MLPs for each pairwise feature interaction $f_{ij}$. Since the numbers of parameters grow quadratically, this setting further exaggerates the parameters and throughput discrepancies. In fact, as we show in Section 4.4, for high dimensional datasets the $NA^2M$ approach does not scale at all.

## 4    Experiments

### 4.1    Datasets

**Tabular datasets.**    We report performance on **CA Housing** [10, 46], **FICO** [22], **CoverType** [8, 16, 20], and **Newsgroups** [32, 43] tabular datasets. We perform one-hot encoding for categorical features, and min-max scaling of features to $[0, 1]$ range. Data is split to have $70/10/20$ ratio for training, validation, and, testing, respectively; except for Newsgroups where test split is fixed, so we only split the train part to $85/15$ ratio for train and validation.

We also report performance on **MIMIC-II** [41, 51], **Credit** [17, 19], **Click** [15], **Epsilon** [21], **Higgs** [3, 26], **Microsoft** [40, 49], **Yahoo** [60], and **Year** [66] tabular datasets. For these datasets, we follow [12, 47] to use the same training, validation, and, testing splits, and to perform the same feature normalization: target encoding for categorical features, quantile transformation with 2000 bins for all features to Gaussian distribution.

Additional details for all tabular datasets are given in Table 1 and Appendix Section A.1.

**Image datasets (classification).**    We experiment with two bird classification datasets: **CUB** [18, 62] and **iNaturalist Birds** [27, 59]. CUB images are annotated with keypoint locations of 15 bird parts, and each location is associated with one or more part-attribute labels. iNaturalist Birds contains significantly more bird classes and more challenging scenes compared to CUB, but lacks keypoint annotations. We perform the concept-bottleneck-style [29] interpretable pre-processing: (i) Images are randomly cropped and resized to $448\times448$ size, and passed through a ResNet50 model [25] until the last pooling layer to extract 2048-D features on the $14\times14$ spatial grid. (ii) Part-attribute linear classifiers are trained on the extracted features using part locations and part-attribute labels from CUB training split. (iii) Max-pooling of part-attribute scores over spatial locations is performed to extract 278 interpretable features (*e.g.*, orange legs, striped wings, needle-shaped beak) for each image in both CUB and iNaturalist, using the part-attribute classifiers trained on CUB only. Splits are preset and results are reported on the validation split, which is a common practice in the computer vision community. Additional details are given in Table 1 and Appendix Section A.1.

Table 1: Datasets overview.

| Dataset | Task | #Train | #Val | #Test | Sparse | #Feat | #Class |
|---|---|---|---|---|---|---|---|
| **Tabular dataset** | | | | | | | |
| CA Housing | Regression | 14,447 | 2,065 | 4,128 | No | 8 | – |
| FICO | Binary | 7,321 | 1,046 | 2,092 | No | 39 | 2 |
| CoverType | Multi-class | 406,707 | 58,102 | 116,203 | No | 54 | 7 |
| Newsgroups | Multi-class | 9,899 | 1,415 | 7,532 | 99.9% | 146,016 | 20 |
| MIMIC-II | Binary | 17,155 | 2,451 | 4,902 | No | 17 | 2 |
| Credit | Binary | 199,364 | 28,481 | 56,962 | No | 30 | 2 |
| Click | Binary | 800,000 | 100,000 | 100,000 | No | 11 | 2 |
| Epsilon | Binary | 320,000 | 80,000 | 100,000 | No | 2,000 | 2 |
| Higgs | Binary | 8,400,000 | 2,100,000 | 500,000 | No | 28 | 2 |
| Microsoft | Regression | 580,539 | 142,873 | 241,521 | No | 136 | – |
| Yahoo | Regression | 473,134 | 71,083 | 165,660 | No | 699 | – |
| Year | Regression | 370,972 | 92,743 | 51,630 | No | 90 | – |
| **Image dataset** | | | | | | | |
| CUB | Classification | 5,994 | 5,794 | – | No | 278 | 200 |
| iNaturalist Birds | Classification | 414,847 | 14,860 | – | No | 278 | 1,486 |
| Common Objects | Detection | 2,645,488 | 58,525 | – | 97.0% | 2,618 | 115 |

**Image dataset (object detection).** For this task we use a proprietary object detection dataset, denoted as **Common Objects**, that contains 114 common objects, (plus a background class) with bounding box locations, 200 parts and 54 attributes. Each box for each image is pre-processed using compositions of parts and attributes to extract 2,618 interpretable features and 100k pairwise feature interactions. For the purpose of evaluating models in this paper, one data instance in this dataset is equivalent to one bounding box. As with previous computer vision datasets, results are reported on the validation split. Additional details are given in Table 1 and Appendix Section A.1.

## 4.2 Baselines

We implement the following baselines in PyTorch [48], and train using mini-batch gradient descent:

**Linear.** Linear / logistic regression are interpretable models that make a prediction decision based on the value of a linear combination of the features.

**NAM [2].** We experiment with two proposed NAM architectures [2]: (i) MLPs containing 3 hidden layers with 64, 64 and 32 units and ReLU [23] activation, and (ii) single hidden layer MLPs with 1,024 ExU units and ReLU-1 activation. We reimplement the original NAM implementation [42] (details in Appendix Section A.3) achieving around $\times 2$–$\times 10$ speedup at training, depending on the dataset. Finally, we extend the implementation to $NA^2Ms$, as well. We released our NAM implementation together with the rest of our code at `github.com/facebookresearch/nbm-spam`.

**MLP.** Multi-layer perceptron (MLP) is a non-interpretable black-box model capturing high-order interaction between the features: $g(\boldsymbol{x}) = f(x_1, x_2, \ldots, x_D)$. For most datasets, MLP sets the upper bound on the performance, and gives an idea of the trade-off between accuracy and interpretability. We experiment with the following architectures: (i) 5 hidden layers with 128, 128, 64, 64, 64 units; (ii) 3 hidden layers with 1024, 512, 512 units; (iii) 2 hidden layers with 2048, 1024 units. We have observed that increasing the depth by adding more layers for any of the three architectures has no additional accuracy gain. For all datasets, we report the best performing architecture across the three. For the following baselines, we use the available implementations:

**EBM [36, 37].** Explainable Boosting Machines (EBMs) are another state-of-the-art GAM which use gradient boosting of millions of shallow trees to learn a shape function for each input feature. These models support automatic pairwise interactions through their $EB^2M$ implementation, but only for regression and binary classification tasks. We use the `interpretml` library [44].

**XGBoost [13].** EXtreme Gradient Boosted trees (XGBoost) are another non-interpretable black-box model that learn high-order feature interactions. We use the `xgboost` library [13].

Table 2: Number of parameters (#par.) and throughput as inputs per second ($x$/sec). Here NAM and NA$^2$M refers to our optimized implementation; $^\dagger$refers to sparse NBM optimization.

| Model | CA Housing | | FICO | | CoverType | | Newsgroups | | iNat. Birds | |
|---|---|---|---|---|---|---|---|---|---|---|
| | #par. | $x$/sec | #par. | $x$/sec | #par. | $x$/sec | #par. | $x$/sec | #par. | $x$/sec |
| NAM | 54K | 0.5M | 262K | 123K | 363K | 80K | 984M | 23 | 2.3M | 15K |
| NBM | 65K | 3.4M$_{\times 6.8}$ | 68K | 821K$_{\times 6.7}$ | 70K | 530K$_{\times 6.6}$ | 18M | $^\dagger$9K$_{\times 391}$ | 0.5M | 74K$_{\times 4.9}$ |
| NA$^2$M | 243K | 119K | 5.3M | 6K | 10M | 3K | – | – | 320M | 99 |
| NB$^2$M | 161K | 641K$_{\times 5.4}$ | 0.3M | 30K$_{\times 5.0}$ | 0.5M | 15K$_{\times 5.0}$ | – | – | 66M | 374$_{\times 3.8}$ |

## 4.3 Implementation details

**NBM.** We use the following architecture for NBMs and NB$^2$Ms: MLP containing 3 hidden layers with 256, 128, and 128 units, ReLU [23], $B = 100$ basis outputs for NBMs and $B = 200$ for NB$^2$Ms. Source code is available at `github.com/facebookresearch/nbm-spam`.

**Training details.** Linear, NAM, NBM, and MLP models are trained using the Adam with decoupled weight decay (AdamW) optimizer [35], on 8×V100 GPU machines with 32 GB memory, and a batch size of at most 1024 per GPU (divided by 2 every time a batch cannot fit in the memory). We train for 1,000, 500, 100, or, 50 epochs, depending on the size and feature dimensionality of the dataset. The learning rate is decayed with cosine annealing [34] from the starting value until zero. For NBMs on all datasets, we tune the starting learning rate in the continuous interval $[1e{-}5, 1.0)$, weight decay in the interval $[1e{-}10, 1.0)$, output penalty coefficient in the interval $[1e{-}7, 100)$, dropout and basis dropout coefficients in the discrete set $\{0, 0.05, 0.1, 0.2, 0.3, 0.4, 0.5, 0.6, 0.7, 0.8, 0.9\}$. We find optimal hyper-parameters using validation set and random search. Similar hyper-parameter search is performed for Linear, NAM, and MLP baselines. Finally, for EBMs and XGBoost, CPU machines are used, with hyper-parameter search using the guidelines in the original works. See Appendix Section A.2 for more training and hyper-parameter search details.

**Evaluation details.** Performance metrics: (i) for regression we report mean-squared error (MSE) or root MSE (RMSE); (ii) for binary classification we report area under the ROC curve (AUROC) or error rate (Error); (iii) for multi-class classification in tabular and image domains we report accuracy@1 (acc@1); and, (iv) for object detection in image domain we report mean average precision (mAP) averaged over IoU thresholds from 0.5 to 0.9 as per MS-COCO [33] definitions. We report average performance and standard deviation over 10 runs with different random seeds.

## 4.4 Comparison with baselines

In this section we compare NBM, as well as the pairwise interactions NB$^2$M counterpart, with popular and widely used GAMs and non-interpretable black-box models.

First, we perform extensive comparison on the number of parameters and throughput against the most similar architecture, *i.e.* Neural Additive Model (NAM) [2]. Both NAMs and NBMs utilize deep networks and are able to run on GPUs, which helps to scale the models on large datasets. Results on representative datasets are presented in Table 2. The throughput is measured as the number of input instances that we can process per second ($x$ / sec) on one 32 GB V100 GPU, in inference mode. For each model, we take the largest batch size (up to 8,192) that fits on the GPU and calculate the average time over 100 runs to process that batch. With that, we calculate the number of input instances processed per second. Throughput can vary depending on the implementation, library used, *etc*. Hence, in order to be as fair as possible, we compare both models with our own optimized implementation in the same deep learning library, *i.e.*, PyTorch [48]. The only dataset where NAM narrowly beats NBM in the number of parameters is CA Housing, which has the smallest number of input features $D = 8$. On other datasets, NBM has around 5×–50× fewer parameters than NAM, while having 4×–7× smaller runtime. Finally, only the sparse version of NBM model can efficiently run on Newsgroups and Common Objects (with pairwise interactions) datasets, where standard NAMs (and other GAMs) have throughput too low for any practical application. Note that these comparisons are with our optimized implementation of NAM.

Table 3: Performance comparison with baselines. ↓: lower is better; ↑: higher is better.

| Model | CA Housing | FICO | CoverType | News. | CUB | iNat. Birds | Common Objects |
|---|---|---|---|---|---|---|---|
| | RMSE ↓ | AUROC ↑ | acc@1 ↑ | acc@1 ↑ | acc@1 ↑ | acc@1 ↑ | mAP ↑ |
| Linear | 0.7354 ±0.0004 | 0.7909 ±0.0001 | 0.7254 ±0.0000 | 0.8238 ±0.0005 | 0.7451 ±0.0003 | 0.3932 ±0.0004 | 0.1917 ±0.0001 |
| EBM | 0.5586 ±0.0002 | 0.7985 ±0.0001 | 0.7392 ±0.0004 | — | — | — | — |
| NAM | 0.5721 ±0.0054 | 0.7993 ±0.0004 | 0.7359 ±0.0003 | | 0.7632 ±0.0010 | 0.4194 ±0.0007 | 0.2056 ±0.0005 |
| NBM | 0.5638 ±0.0013 | 0.8048 ±0.0005 | 0.7369 ±0.0002 | 0.8446 ±0.0012 | 0.7683 ±0.0007 | 0.4227 ±0.0007 | 0.2168 ±0.0006 |
| $EB^2M$ | 0.4919 ±0.0004 | 0.7998 ±0.0005 | — | — | — | — | — |
| $NA^2M$ | 0.4921 ±0.0078 | 0.7992 ±0.0003 | 0.8872 ±0.0006 | | 0.7713 ±0.0011 | 0.4591 ±0.0010 | — |
| $NB^2M$ | 0.4779 ±0.0020 | 0.8029 ±0.0003 | 0.8908 ±0.0008 | | 0.7770 ±0.0006 | 0.4684 ±0.0009 | 0.2378 ±0.0006 |
| XGBoost | 0.4428 ±0.0006 | 0.7925 ±0.0008 | 0.8860 ±0.0003 | 0.7677 ±0.0009 | 0.7186 ±0.0008 | — | — |
| MLP | 0.5014 ±0.0061 | 0.7936 ±0.0013 | 0.9694 ±0.0002 | 0.8494 ±0.0021 | 0.7684 ±0.0007 | 0.4584 ±0.0008 | 0.2376 ±0.0007 |

Next, we compare performance with GAM baselines, and non-interpretable black-box models in Table 3. We notice that NBMs achieve the best performance among all GAMs, even outperforming full complexity MLPs on several datasets. The difference in performance is more pronounced on larger datasets, such as iNaturalist and Common Objects, where NBMs generalize better. EBMs have a big downside as they do not support pairwise interactions on multi-class tasks, and do not scale at all to very large datasets. Finally, on datasets with large number of features, *i.e.* Newsgroups and Common Objects (with pairwise interactions), NBMs are the only GAMs that scale, as we did not manage to complete a successful training with NAMs or EBMs.

We finally observe that pairwise interactions are enough to match or beat black-box models on 6 out of 7 datasets. The only exception is CoverType, where MLP has 0.9694 acc@1 *vs.* $NB^2M$ with 0.8908. However, we scale NBMs further and train $NB^3M$ (*i.e.*, including triplet interactions) that gets 0.9634 acc@1. Even though this is an interesting result that helps us understand the data behavior, we argue that triplet feature interactions are very hard to visualize and hence *not* interpretable.

## 4.5 Comparison with state of the art

We finally compare NBM, as well as the pairwise interactions $NB^2M$ counterpart, with neural-based state-of-the-art GAMs. Namely, we compare with Neural Additive Model (NAM) [2] and Neural Oblivious Decision Trees GAM (NODE-GAM) [12], and their pairwise interactions versions $NA^2M$ and NODE-GA$^2$M. Results are presented in Table 4.

We compute NAM and NBM results using our codebase, while NODE-GAM results are reproduced from [12]. For a fair comparison on these datasets, we use training, validation, and, testing splits from NODE-GAM [12], and perform the same feature normalization, see Section 4.1 for details. Without any additional model selection (architecture tuning, stochastic weight averaging, *etc.*) NBMs outperform NODE-GAMs on 7 out of 8 presented datasets (albeit marginally), and outperform NAM on all datasets. Note that, the idea of NBM is perpendicular to that of NODE-GAM and it is possible that bigger gains can be achieved by combining them, especially for Epsilon and Yahoo datasets, where $NB^2M$s do not scale due to a high dimensionality and non-sparse nature of features.

Table 4: Performance comparison with state-of-the-art GAMs. NODE-GAM results are reproduced from [12]. ↓: lower is better; ↑: higher is better. State-of-the-art performance shown in **bold**.

| Model | MIMIC-II | Credit | Click | Epsilon | Higgs | Microsoft | Yahoo | Year |
|---|---|---|---|---|---|---|---|---|
| | AUROC ↑ | AUROC ↑ | Error ↓ | Error ↓ | Error ↓ | MSE ↓ | MSE ↓ | MSE ↓ |
| NAM | 0.8539 ±0.0004 | 0.9766 ±0.0027 | 0.3317 ±0.0005 | 0.1079 ±0.0002 | 0.2972 ±0.0001 | 0.5824 ±0.0002 | 0.6093 ±0.0003 | 85.25 ±0.01 |
| NODE GAM | 0.8320 ±0.0110 | 0.9810 ±0.0110 | 0.3342 ±0.0001 | 0.1040 ±0.0003 | 0.2970 ±0.0001 | 0.5821 ±0.0004 | 0.6101 ±0.0006 | **85.09** ±0.01 |
| NBM | **0.8549** ±0.0004 | **0.9829** ±0.0014 | **0.3312** ±0.0002 | **0.1038** ±0.0002 | **0.2969** ±0.0001 | **0.5817** ±0.0001 | **0.6084** ±0.0001 | 85.10 ±0.01 |
| NA²M | 0.8639 ±0.0011 | 0.9824 ±0.0032 | 0.3290 ±0.0005 | — | 0.2555 ±0.0003 | 0.5622 ±0.0003 | — | 79.80 ±0.05 |
| NODE GA²M | 0.8460 ±0.0110 | **0.9860** ±0.0100 | 0.3307 ±0.0001 | 0.1050 ±0.0002 | 0.2566 ±0.0003 | **0.5618** ±0.0003 | **0.5807** ±0.0004 | 79.57 ±0.12 |
| NB²M | **0.8690** ±0.0010 | 0.9856 ±0.0017 | **0.3286** ±0.0002 | — | **0.2545** ±0.0002 | **0.5618** ±0.0002 | — | **79.01** ±0.03 |

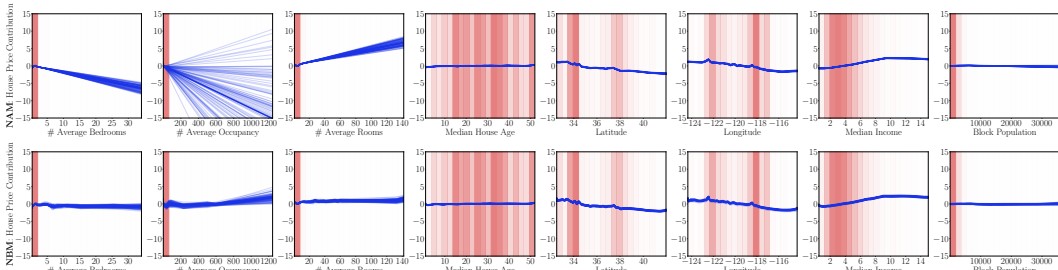

Figure 3: NAM (upper row) and NBM (bottom row) shape functions $f_i$ for the CA Housing dataset.

## 4.6 Stability and interpretability

The interpretability of GAMs comes from the fact that the learned shape functions can be easily visualized. In the same manner as the other GAM approaches, each feature's importance in an NBM can be represented by a unique shape function that *exactly* describes how the NBM computes a prediction. We demonstrate this on the CA Housing dataset because it has a small number of input features ($D = 8$) and the full model can conveniently be visualized in a single row, see Figure 3.

Towards this purpose, we train an ensemble of 100 models by running different random seeds with optimal hyper-parameters, in order to analyze when the models learn the same shape and when they diverge. Following Agarwal et al. [2], we set the average score for each shape function to be zero by subtracting the respective mean feature score. Next, we plot each shape function as $f_i(x_i)$ *vs.* $x_i$ for each model in the ensemble using a semi-transparent line, and an average ensemble shape function using a thick line. Finally, the $x$-axis is divided by bars depicting the normalized data density, *i.e.*, darker areas contain more data points. Figure 3 depicts an ensemble of NAMs [2] (upper row) and NBMs (bottom row). Although the shape functions are correlated for most features, we observe that NBMs diverge much less in the cases where there are only few data points (light / white areas in the graphs). This is due to the fact that the basis network in NBM is trained jointly with only linear composing weights being trained for each feature separately. In contrast, each feature in NAM has its own network, which becomes unstable and diverges in cases of few training data points.

Finally, to quantify stability, we compute the standard deviation for each visualized shape function over 100 models, and report the mean standard deviation over all features. For NAMs, this mean standard deviation is 0.9921, while for NBMs it is 0.1987.

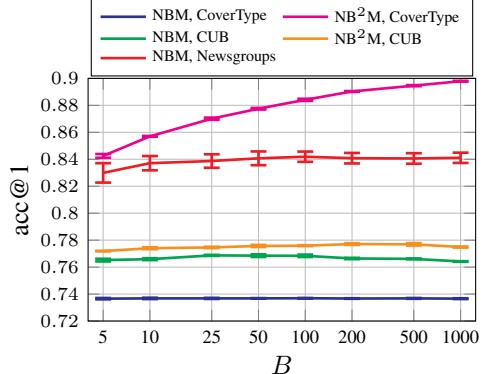

Figure 4: Ablation on the number of bases $B$.

Table 5: Ablation on the number of subnets $S$.

| Model | $S$ | CoverType | | |
| --- | --- | --- | --- | --- |
| | | #param | $x$/sec | acc@1 |
| NAM | 1 | 363K | 80K | 0.7359 |
| NBM | 1 | 70K | 530K | 0.7369 |
| NAM | 5 | 1.8M | 16K | 0.7417 |
| NBM | 5 | 350K | 88K | 0.7435 |
| NB$^2$M | 1 | 463K | 15K | 0.8908 |

## 4.7 Ablation study

**Number of basis functions.** We evaluate the robustness of NBMs *w.r.t.* the choice of the number of basis functions $B$. Results are presented in Figure 4 for Newsgroups, CoverType, and, CUB. We observe that NBMs are not overly sensitive to the choice of $B$. Rather than tuning this hyperparameter, we recommend setting $B = 100$ for NBMs and $B = 200$ for NB$^2$Ms as it performs well across a large variety of datasets we experimented with. Although, *e.g.*, for NB$^2$Ms on CoverType, using a larger number of basis functions leads to some performance gains, it comes with a throughput trade-off, as computing the linear combination of bases starts becoming the bottleneck. Thus, we suggest using the recommended values as a great trade-off between accuracy and throughput.

**Multiple subnets.** Agarwal et al. [2] propose to extend NAMs to multi-task / multi-class setups by associating each feature with multiple subnets. This method is directly applicable on NBMs as well, by using $S$ different networks to learn $S$ sets of basis functions. We compare few options on the multi-class CoverType predictions with the number of subnets $S$=1 and $S$=5 in Table 5. Both NAMs and NBMs have a similarly small relative accuracy improvement when using 5 subnets over the 1 subnet, however that comes at a $5\times$ increase in the number of parameters and $5\times$ lower throughput. With the accuracy-complexity trade-off in mind, we did not see much benefit of using $S = 5$ on our datasets, so we keep $S = 1$ across all other experiments. Interestingly, NB$^2$M with $S$=1 has a comparable throughput to NAM with $S$=5, while achieving an impressive accuracy gain, see Table 5.

## 5 Conclusion and future work

This work describes novel Neural Basis Models (NBMs), which belong to a new subfamily of Generalized Additive Models (GAMs), and utilize deep learning techniques to scale to large datasets and high-dimensional features. Our approach addresses several scalability and performance issues associated with GAMs, while still preserving their interpretability compared to black-box deep neural networks (DNNs). We show that our NBMs and NB$^2$Ms achieve state-of-the-art performance on a large variety of datasets and tasks, while being much smaller and faster than other neural-based GAMs. As a result, they can be used as a drop-in replacements for the black-box DNNs.

We do recognize that our approach, though highly scalable, has limitations *w.r.t.* number of features. Beyond 10,000 dense (or 1 million sparse) features, we would need to apply some form of feature selection [12, 37, 58] to scale further. Scalability issue is even more pronounced when modeling pairwise interactions in NB$^2$M. However, NBMs can still handle an order of magnitude more than what can be handled by NAM or other GAM approaches that do not perform feature selection.

There are many future directions for this line of work. First, for the computer vision domain, we assume an intermediate, interpretable concept layer [29] on which NBMs and in general GAMs can be applied. It would be interesting to explore visual interpretability by either directly going to the pixel space with NBMs or learning visual features that can do recognition with lower order interactions in NBM framework (for example, NB$^2$Ms). Finally, the idea of using shared basis is generic. Although, we used neural networks to learn these basis, we can enhance the model interpretability further for higher order interactions, by using more interpretable learning functions such as polynomials.

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
