# A  Appendix

## A.1  Dataset descriptions

**CA Housing [10, 46].**    The target to **regress** is the median house value for California districts (1990 U.S. census), expressed in hundreds of thousands of dollars, from 8 household-related variables.

**FICO [22].**    Anonymized dataset of line of credit applications made by real homeowners. The customers in this dataset have requested a credit line in the range of $5,000 – $150,000. The task is to make a **binary prediction** whether applicants will repay their account within 2 years.

**CoverType [8, 16, 20].**    The samples in this dataset correspond to 30×30m patches of forest in the U.S., collected for the task of predicting each patch's cover type, *i.e.*, the dominant species of tree. There are seven covertypes, making this a **multi-class classification** problem.

**Newsgroups [32, 43].**    This dataset is a collection of news documents, partitioned across 20 different newsgroups, making this a **multi-class classification** problem. Each article is represented by a *tf-idf* term for each word in the training split word vocabulary. Newsgroups dataset has sparsity 99.9%, *i.e.*, only around 150 words from a vocabulary of 150k words appears per a given article, on average.

**MIMIC-II [41, 51].**    The task is to make a **binary prediction** on mortality of Intensive Care Unit (ICU) patients. Contains physiologic signals and vital signs time series captured from patient monitors for tens of thousands of ICU patients.

**Credit [17, 19].**    This dataset contains anonymized credit card transactions labeled as fraudulent or genuine, and the task is to make a **binary prediction** between them.

**Click [15].**    Subset of data from the KDD Cup 2012. Namely, 500,000 objects of a positive class and 500,000 objects of a negative class were randomly sampled to create a **binary prediction** task. We would like to gratefully acknowledge the organizers of KDD Cup 2012 as well as Tencent Inc. for making the dataset available.

**Epsilon [21].**    Dataset for **binary prediction** from the PASCAL Large Scale Learning Challenge.

**Higgs [3, 26].**    The problem is to **binary predict** whether the given event produces Higgs bosons.

**Microsoft [40, 49].**    Ranking dataset, where features are extracted from query-url pairs. Each pair has relevance judgment labels to **regress**, which take values from 0 (irrelevant) to 4 (very relevant).

**Yahoo [60].**    Another ranking dataset with query-url pairs that have labels to **regress** from 0 to 4.

**Year [66].**    Subset of Million Song Dataset. The task is to **regress** the release year of the song by using the audio features. It contains tracks from 1922 to 2011.

**CUB [18, 62].**    This **image classification** dataset consists of images of 200 bird classes. All images are annotated with keypoint locations of 15 bird parts (*e.g.*, beak, wing, crown) and each location is associated with one or more part-attribute labels (*e.g.*, orange leg, striped wing). Some of the keypoint annotations distinguish between the left-right instances of parts, *e.g.*, 'left wing' / 'right wing', 'left eye' / 'right eye'. We treat these as the same part, *i.e.*, 'left wing' and 'right wing' as 'wing'.

**iNaturalist Birds [27, 59].**    Another **image classification** dataset that contains 1,486 bird classes. The full iNaturalist 2021 dataset consists of various super-categories (*e.g.*, plants, insects, birds), covering 10K species in total. The Birds super-category contains 1,486 bird classes and more challenging scenes compared to CUB. Therefore, the iNaturalist dataset is a challenging testbed for any image classification method. However, note that this dataset lacks keypoint annotations.

**Common Objects.**    Proprietary **object detection** dataset created by collecting public images from Instagram[1]. Dataset contains 114 common household objects, (*e.g.*, stove, bed, table), plus a background class, with bounding box locations, 200 parts and 54 attributes. Each bounding box for each image is pre-processed using compositions of parts (*e.g.*, leg, handle, top) and attributes (*e.g.*, colors, textures, shapes) to extract 2,618 interpretable features and 100k pairwise feature interactions. Common Objects dataset has sparsity 97%, *i.e.*, only around 76 part-attribute compositions from a vocabulary of 2618 compositions are active for a given object, on average.

---

[1] www.instagram.com

Table A.1: Optimal hyper-parameters for NBMs and NB$^2$Ms on all datasets.

**NBM:** $[256, 256, 128]$ hidden units, 100 basis functions

| Dataset | Number of epochs | Batch size | Learning rate | Weight decay | Dropout | Basis dropout | Output penalty |
|---|---|---|---|---|---|---|---|
| **CA Housing** | 1,000 | 1,024 | 0.00197 | 1.568e-5 | 0.0 | 0.05 | 1.439e-4 |
| **FICO** | 1,000 | 1,024 | 0.02176 | 1.684e-5 | 0.3 | 0.7 | 2.462e-4 |
| **CoverType** | 500 | 1,024 | 0.01990 | 5.931e-7 | 0.0 | 0.0 | 0.05533 |
| **Newsgroups** | 500 | 512 | 3.133e-4 | 1.593e-8 | 0.1 | 0.3 | 4.578 |
| **MIMIC-II** | 1,000 | 1,024 | 0.01460 | 3.177e-6 | 0.5 | 0.1 | 2.318 |
| **Credit** | 500 | 1,024 | 0.00391 | 1.574e-6 | 0.0 | 0.9 | 0.03737 |
| **Click** | 500 | 1,024 | 2.745e-4 | 7.21e-10 | 0.0 | 0.5 | 20.085 |
| **Epsilon** | 500 | 1,024 | 3.776e-5 | 1.507e-7 | 0.3 | 0.4 | 0.00273 |
| **Higgs** | 50 | 1,024 | 1.792e-4 | 1.087e-9 | 0.0 | 0.0 | 3.906e-5 |
| **Microsoft** | 500 | 1,024 | 1.677e-4 | 1.969e-7 | 0.1 | 0.3 | 1.986e-4 |
| **Yahoo** | 500 | 1,024 | 0.00446 | 1.399e-8 | 0.1 | 0.3 | 0.01688 |
| **Year** | 500 | 1,024 | 8.780e-5 | 1.580e-7 | 0.1 | 0.1 | 2.592e-5 |
| **CUB** | 500 | 128 | 0.01173 | 0.12910 | 0.7 | 0.3 | 4.739 |
| **iNaturalist Birds** | 100 | 1,024 | 0.00140 | 3.548e-5 | 0.0 | 0.2 | 1.423e-5 |
| **Common Objects** | 100 | 1,024 | 0.12480 | 1.001e-5 | 0.1 | 0.0 | 0.0 |

**NB$^2$M:** $[256, 256, 128]$ hidden units, 200 basis functions

| Dataset | Number of epochs | Batch size | Learning rate | Weight decay | Dropout | Basis dropout | Output penalty |
|---|---|---|---|---|---|---|---|
| **CA Housing** | 1,000 | 1,024 | 0.00190 | 7.483e-9 | 0.0 | 0.05 | 1.778e-6 |
| **FICO** | 1,000 | 1,024 | 2.287e-4 | 3.546e-7 | 0.1 | 0.7 | 0.19330 |
| **CoverType** | 500 | 512 | 0.00268 | 1.660e-7 | 0.0 | 0.0 | 0.00155 |
| **MIMIC-II** | 1,000 | 1,024 | 1.796e-4 | 3.494e-4 | 0.1 | 0.5 | 0.05964 |
| **Credit** | 500 | 1,024 | 3.745e-4 | 4.610e-5 | 0.5 | 0.0 | 0.25280 |
| **Click** | 500 | 1,024 | 9.614e-4 | 0.00159 | 0.0 | 0.5 | 0.05773 |
| **Higgs** | 50 | 1,024 | 0.00201 | 2.202e-4 | 0.0 | 0.1 | 1.969e-7 |
| **Microsoft** | 100 | 128 | 1.640e-4 | 1.552e-8 | 0.0 | 0.9 | 2.928e-6 |
| **Year** | 100 | 256 | 3.180e-4 | 1.696e-8 | 0.0 | 0.9 | 4.454e-4 |
| **CUB** | 500 | 32 | 2.629e-4 | 0.03209 | 0.0 | 0.0 | 96.894 |
| **iNaturalist Birds** | 100 | 32 | 6.735e-5 | 9.870e-5 | 0.05 | 0.0 | 3.785 |
| **Common Objects** | 100 | 64 | 0.03127 | 1.013e-4 | 0.0 | 0.2 | 8.126 |

## A.2 Hyper-parameters

Linear, MLP, NAM, and NBM are trained using the Adam with decoupled weight decay (AdamW) optimizer [35], on $8 \times$ V100 GPU machines with 32 GB memory, and a batch size of at most 1024 per GPU. We train for 1,000, 500, 100, or, 50 epochs, depending on the size and feature dimensionality of the dataset. The learning rate is decayed with cosine annealing [34] from the starting value until zero. We find optimal hyper-parameters for all models using validation set and random search, following the detailed guidelines.

**Linear.** We tune the starting learning rate in the continuous interval $[1e-5, 100)$, weight decay in the interval $[1e-10, 1.0)$.

**MLP.** We tune the starting learning rate in the continuous interval $[1e-5, 1.0)$, weight decay in the interval $[1e-10, 1.0)$, dropout coefficients in the discrete set $\{0, 0.05, 0.1, 0.2, 0.3, 0.4, 0.5, 0.6, 0.7, 0.8, 0.9\}$.

**NAM.** We tune the starting learning rate in the continuous interval $[1e-5, 1.0)$, weight decay in the interval $[1e-10, 1.0)$, output penalty coefficient in the interval $[1e-7, 100)$, dropout and feature dropout coefficients in the discrete set $\{0, 0.05, 0.1, 0.2, 0.3, 0.4, 0.5, 0.6, 0.7, 0.8, 0.9\}$.

**NBM.** We tune the starting learning rate in the continuous interval $[1e-5, 1.0)$, weight decay in the interval $[1e-10, 1.0)$, output penalty coefficient in the interval $[1e-7, 100)$, dropout and basis dropout coefficients in the discrete set $\{0, 0.05, 0.1, 0.2, 0.3, 0.4, 0.5, 0.6, 0.7, 0.8, 0.9\}$. Optimal hyper-parameters for NBMs and NB$^2$Ms on all datasets are given in Table A.1.

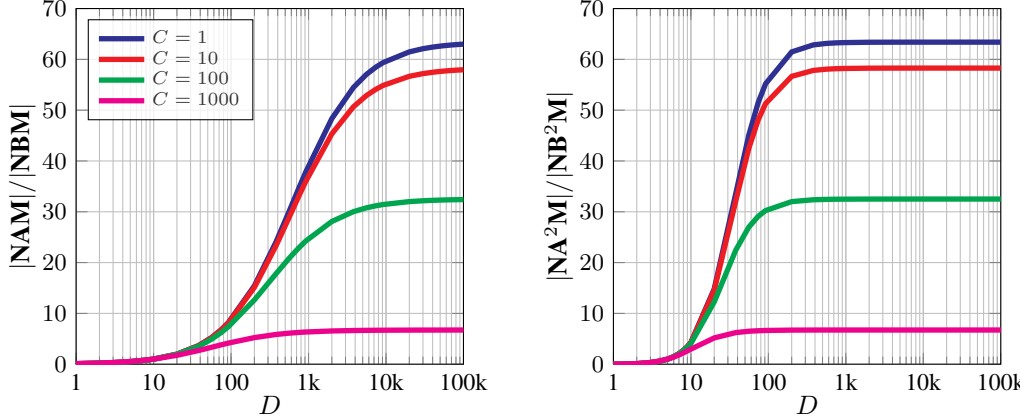

Figure A.1: NAM *vs.* NBM: #parameters (left); NA$^2$M *vs.* NB$^2$M: #parameters (right).

Finally, for EBMs and XGBoost, CPU machines are used, with hyper-parameter search as follows.

**EBM.** We tune the maximum bins from the set $\{8, 16, 32, 64, 128, 256, 512\}$, number of interactions from $\{0, 2, 4, 8, 16, 32, 64, 128, 256, 512\}$ (they are set to 0 for EBMs and $\geq 0$ for EB$^2$Ms), learning rate in the continuous range from $[1e-6, 100)$, the maximum rounds from the set $\{1000, 2000, 4000, 8000, 16000\}$, the minimum samples in a leaf node from the set $\{1, 2, 4, 8, 10, 15, 20, 25, 50\}$, and the same range is used for the maximum leaves parameter. For binning, we search within the set {"quantile", "uniform", "quantile_humanized"}. The inner bags and outer bags are selected from the range $\{1, 2, 4, 8, 16, 32, 64, 128\}$.

**XGBoost.** We tune the number of estimators from $\{1, 2, 4, 8, 10, 20, 50, 100, 200, 250, 500, 1000\}$, the max-depth from the set $\{\infty, 2, 5, 10, 20, 25, 50, 100, 2000\}$, $\eta$ over a continuous range $[0.0, 1.0)$, and use the same for the `subsample` and `colsample_bytree` parameters.

### A.3 Additional discussion *w.r.t.* NAM

**Number of parameters for multi-class and pairwise feature interactions.** We compare number of weight parameters needed to learn NAM *vs.* NBM for the multi-class task. This discussion is an extension of the discussion in Section 3.3. Let us denote with $M$ the number of parameters in MLP for each feature in NAM, and with $N$ the number of parameters in MLP for bases in NBM. In most experiments the optimal NAM has 3 hidden layers with 64, 64 and 32 units ($M = 6401$), and, NBM has 3 hidden layers with 256, 128, 128 units ($N = 62820$) and $B = 100$ basis functions. Finally, let us denote with $D$ the input feature dimensionality, and with $C$ the number of classes in the multi-class task. Then the ratio of number of parameters in NAM *vs.* NBM is given by,

$$\frac{|\textbf{NAM}|}{|\textbf{NBM}|} = \frac{D \cdot M + D \cdot C}{N + D \cdot B + D \cdot C} = \frac{6401 + C}{\frac{62820}{D} + 100 + C}. \tag{A.1}$$

Similarly, in the case of pairwise feature interactions in NA$^2$M *vs.* NB$^2$M, this ratio is given by,

$$\frac{|\textbf{NA}^2\textbf{M}|}{|\textbf{NB}^2\textbf{M}|} = \frac{\frac{D(D-1)}{2} \cdot M + \frac{D(D-1)}{2} \cdot C}{N + \frac{D(D-1)}{2} \cdot B + \frac{D(D-1)}{2} \cdot C} = \frac{6401 + C}{\frac{125640}{D(D-1)} + 100 + C}. \tag{A.2}$$

Figure A.1 shows both ratios for different values of feature dimensionality $D$ and number of classes $C$.

For unary features (NAM *vs.* NBM, Figure A.1 left), the conclusion is the same as in the case of binary classification, *i.e.*, for $D = 10$, NBMs and NAMs have roughly equal number of parameters, for any given value of $C$. For higher number of classes, NBMs still provide significant gain over NAMs, however, that gain starts decreasing, due to the fact that the final linear classifier starts becoming the most memory hungry part of the model. Nevertheless, even with $C = 1,486$ and $D = 278$ in iNaturalist Birds, which has the most classes from the datasets we used, NBMs have around 5× less parameters than NAMs, see Table 2.

For pairwise feature interactions (NA$^2$M *vs.* NB$^2$M, Figure A.1 right), the ratio is much more pronounced, *i.e.*, already at $D = 5$, NB$^2$Ms and NA$^2$Ms have equal number of parameters, and the growth of the ratio *w.r.t.* $D$ is much more significant. Already after few hundred feature dimensions the ratio is at peak.

**Throughput optimization of NAMs.** Neural Additive Models (NAMs) [2] learn an MLP network for each input feature, followed by a linear combination to make a prediction. Official implementation [42] runs a `for loop` over all networks, which results in a poor GPU utilization. More precisely, this implementation requires extremely large batches ($\gg$1,024) per GPU to make the training efficient, which is impractical. We do recognize that efficiency was not of highest priority to the authors [2], but in our case we are scaling GAMs to multi-class datasets with order of million data points. Thus, to facilitate a fair comparison against our NBMs, we reimplement NAMs using grouped convolutions [30, 65], which are readily available in standard deep learning libraries. Namely, we stack corresponding hidden layers of all MLPs (*e.g.*, first hidden layer of all MLPs) into a grouped 1-D convolution, where number of groups equals the number of features. The computation performed is identical to original NAMs, *i.e.* there is no feature interaction, while achieving around $\times 2$–$\times 10$ speedup, depending on the dataset. We perform the same implementation trick to NA$^2$Ms, as well.

## A.4 Additional visualization

The interpretability of GAMs comes from the fact that the learned shape functions can be easily visualized. In the same manner as the other GAM approaches, each feature's importance in an NBM can be represented by a unique shape function that *exactly* describes how the NBM computes a prediction. For an example, see Figure 3 visualization on the CA Housing dataset. We additionally demonstrate this on the CUB image classification dataset that consists of 200 bird classes, where each image is represented by interpretable features, *e.g.*, a "bird" image can be represented with "striped wings", "needle-shaped beak", "long legs", *etc.*, that are predicted from the image using a convolutional-neural-network (CNN) model. We present visualizations of shape functions with highest positive or negative contribution to 6 randomly selected bird classes, see Figures A.2 and A.3.

Towards this purpose, we train an ensemble of 20 models by running different random seeds with optimal hyperparameters, in order to analyze when the models learn the same shape and when they diverge. Following Agarwal et al. [2], we set the average score for each shape function to be zero by subtracting the respective mean feature score. Next, we plot each shape function as $f_i(x_i)$ *vs.* $x_i$ for each model in the ensemble using a semi-transparent line, and an average ensemble shape function using a thick line. Finally, the $x$-axis is divided by bars depicting the normalized data density, *i.e.*, darker areas contain more data points. Figures A.2 and A.3 depict few image examples for the respective class (upper row), and an ensemble of NBMs (bottom row). We visually observe that NBMs provide a strong interpretable overview of the respective bird class, and that the shape functions do not diverge significantly even in the cases where there are only few data points (light / white areas in the graphs).

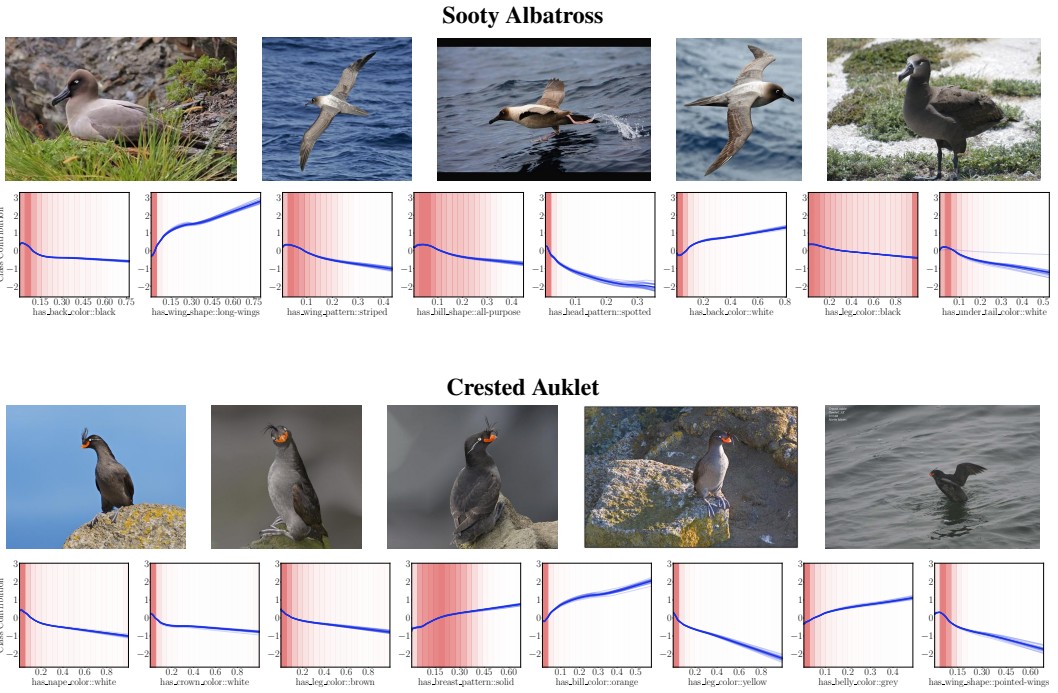

Figure A.2: CUB bird class image examples (upper row) and NBM shape functions $f_i$ (bottom row) with highest positive or negative contribution to the respective bird class prediction.

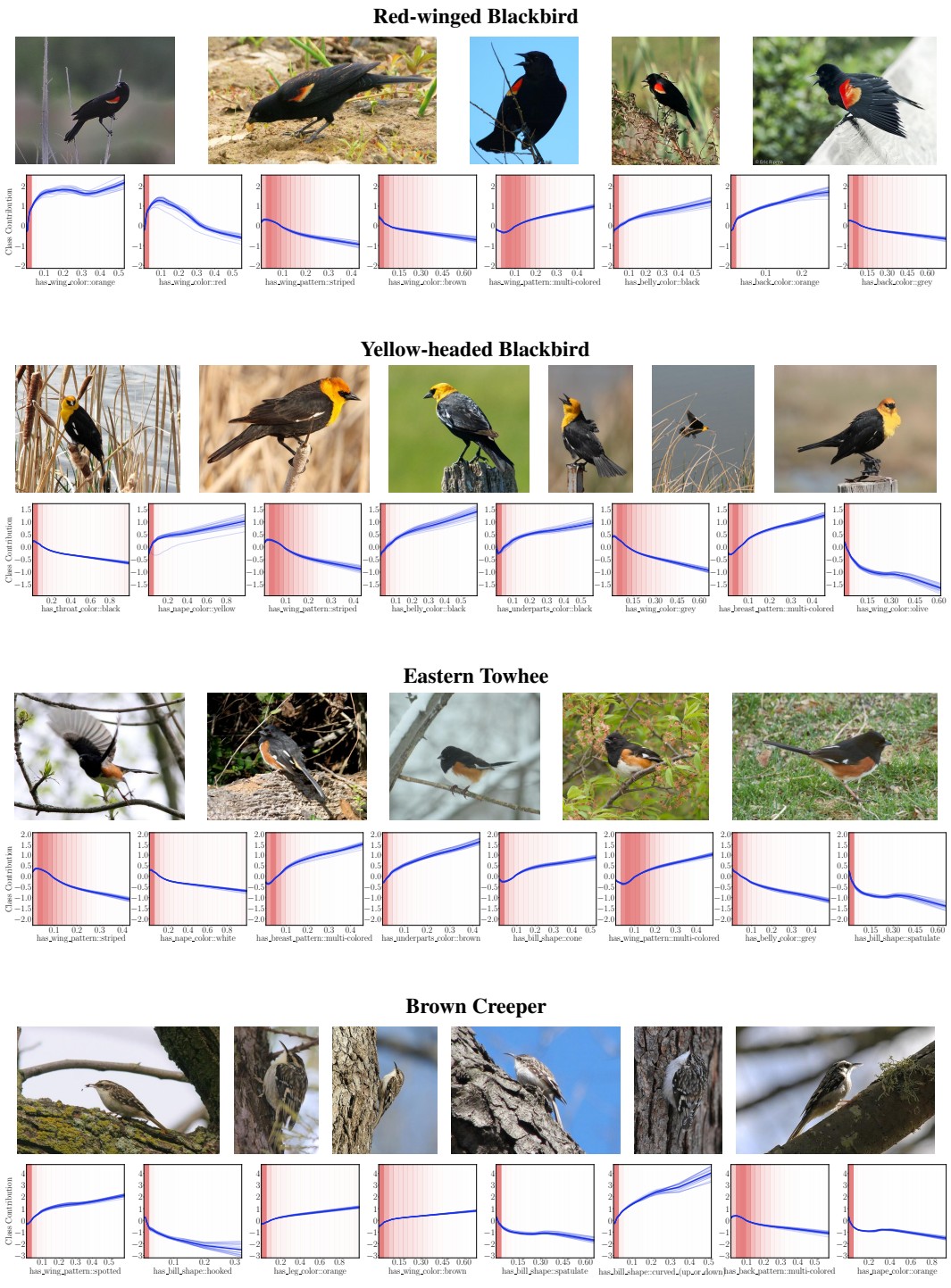

Figure A.3: CUB bird class image examples (upper row) and NBM shape functions $f_i$ (bottom row) with highest positive or negative contribution to the respective bird class prediction.

## A.5 Learning-Theoretic Guarantees for Basis Models in a RKHS

As discussed briefly in the main paper, it is possible to develop a more rigorous argument for the use of a small set of basis functions instead of a complete generalized additive model. To elucidate we first require establishing some notation: We represent matrices by uppercase boldface, *e.g.*, $\mathbf{X}$ and vectors by lowercase boldface, *i.e.*, $\boldsymbol{x}$. We assume that the covariates lie within the set $\mathcal{X} \subseteq \mathbb{R}^D$, and the labels lie within the finite set $\mathcal{Y}$. Data $(\boldsymbol{x}, y) \in \mathcal{X} \times \mathcal{Y}$ are drawn following some unknown (but fixed) distribution $\mathfrak{P}$. We assume we are provided with $n$ i.i.d. samples $\{(\boldsymbol{x}_i, y_i)\}_{i=1}^n$ as the train set.

Consider a generalized additive model (GAM) $g : \mathcal{X} \to \mathcal{Y}$:

$$g(\boldsymbol{x}) = \sum_{i=1}^{D} w_i \cdot f_i(x_i).$$

Assume that the shape functions $f_1, \ldots, f_D; f_i : \mathbb{R} \to \mathcal{Y}$ have a maximum norm $B_{\mathcal{H}} > 0$ in some Reproducing Kernel Hilbert Space (RKHS, [7]) $\mathcal{H}$ endowed with a PSD kernel $k(\cdot, \cdot) : \mathbb{R} \times \mathbb{R} \to \mathbb{R}$ and feature $\boldsymbol{\phi} : \mathbb{R} \to \mathbb{R}^{d_{\mathcal{H}}}$, *i.e.*, $\|f_i\|_{\mathcal{H}} \le B_{\mathcal{H}}$, and $\boldsymbol{w} = \{w_i\}_{i=1}^D \in \mathbb{R}^D$, $k(x, y) = \boldsymbol{\phi}(x)^\top \boldsymbol{\phi}(y)$ such that $\|\boldsymbol{w}\|_2 \le B_{\boldsymbol{w}}$ for $B_{\boldsymbol{w}} > 0$. This characterization corresponds to a family of functions $\mathcal{H}_A$, *i.e.*,

$$\mathcal{H}_A = \{g \mid g(\boldsymbol{x}) = \sum_{i=1}^{D} w_i f_i(x_i), \|f_i\|_{\mathcal{H}} \le B_{\mathcal{H}}, \|\boldsymbol{w}\|_2 \le B_{\boldsymbol{w}}\} \tag{A.3}$$

The idea behind the basis decomposition approach highlighted in this paper is to only use a fixed number of bases, $B$, to model each $f_i$. Observe that one can obtain rigorous guarantees for $f_i$ that lie within an RKHS using Mercer's Theorem [39]. We have that if the kernel $k$ associated with the RKHS $\mathcal{H}$ is continuous, positive-definite and symmetric, there exist a set of eigenvalues $\{\lambda_i\}_{i=1}^\infty$ and eigenfunctions (basis functions) $\{\boldsymbol{\omega}_i\}_{i=1}^\infty$ that form an orthonormal basis for $k$, *i.e.*, for any $x, y \in \mathbb{R}$,

$$k(x, y) = \sum_{i=1}^{\infty} \lambda_i \boldsymbol{\omega}_i(x) \boldsymbol{\omega}_i(y). \tag{A.4}$$

Where the bases are orthonormal, *i.e.*, $\int_{x \in \mathbb{R}} \boldsymbol{\omega}_i(x) \boldsymbol{\omega}_j(x) dx = 0$ for $i \ne j$ and 1 otherwise. This representation naturally gives a form for $\boldsymbol{\phi}(\cdot) = [\sqrt{\lambda_i} \boldsymbol{\omega}_i(\cdot)]_{i=1}^\infty$. Furthermore, we have that for each $f \in \mathcal{H}$ there exists $\boldsymbol{f} \in L^2$ such that $f(x) = \langle \boldsymbol{f}, \boldsymbol{\phi}(x) \rangle_{\mathcal{H}} \forall x \in \mathbb{R}$. Note once again that the reproducing kernel Hilbert space $\mathcal{H}$ corresponds to the feature-wise functions $f$, whereas the space $\mathcal{H}_A$ corresponds to the overall function $g$. Now, we can define, for the family $\mathcal{H}_A$ a **Generalized Basis Model** of order $B$ (denoted as $\mathcal{H}_B$) as the following.

**Definition 1.** A Generalized Basis Model of order $B$ for any function class $\mathcal{H}_A$ that satisfies the characterization in Equation A.3 for some $(\mathcal{H}, B_{\mathcal{H}}, B_{\boldsymbol{w}})$ is given by the family $\mathcal{H}_B$:

$$\mathcal{H}_B = \left\{ g \left| \begin{array}{c} g(\boldsymbol{x}) = \sum_{i=1}^{D} w_i f_i(x_i), \\ f_i(\cdot) = \sum_{j=1}^{B} \beta_{ij} h_j(\cdot), \|f_i\|_{\mathcal{H}} \le B_{\mathcal{H}}, \|\boldsymbol{w}\|_2 \le B_{\boldsymbol{w}}, \\ h_i \in \mathcal{H}, h_i \perp h_j \forall i \ne j. \end{array} \right. \right\}$$

Where orthogonality ($\perp$) is defined as $h_i \perp h_j \implies \int_{x \in \mathbb{R}} h_i(x) \cdot h_j(x) dx = 0$.

Next, note that by Mercer's Theorem, for each function $f \in \mathcal{H}$, there exists $\boldsymbol{f} = \{f_i\}_{i=1}^\infty, \boldsymbol{f} \in L^2$ such that $f(x) = \langle \boldsymbol{f}, \boldsymbol{\phi}(x) \rangle_{\mathcal{H}}$. Combining this statement with the basis representation for $\boldsymbol{\phi}$ gives us an alternate representation of any $f \in \mathcal{H}$, as

$$f(\cdot) = \sum_{i=1}^{\infty} f_i \sqrt{\lambda_i} \boldsymbol{\omega}_i(\cdot).$$

Under this representation, we can relate the two spaces $\mathcal{H}_A$ and $\mathcal{H}_B$ as follows.

**Proposition 1.** *For any $\mathcal{H}$, dimensionality $D$, and number of basis functions $B > 0$, $\mathcal{H}_B \subseteq \mathcal{H}_A$.*

*Proof.* Follows from Mercer's Theorem [39]. Any $g \in \mathcal{H}_B$ can be written as a linear combination of functions in $\mathcal{H}$ (and consequently $\mathcal{H}_A$), each of which admit a basis representation via Mercer's Theorem, where all but $B$ components have coefficient 0. In the limit $B \to \infty$, $\mathcal{H}_B = \mathcal{H}_A$. $\square$

Since the basis functions in $\mathcal{H}_B$ lie on a finite-dimensional subspace within $\mathcal{H}$ spanned by $B$ basis vectors, we can without loss of generality, assume that these $B$ basis vectors correspond to $\{\boldsymbol{\omega}_i\}_{i=1}^B$ obtained from Equation A.4. Now, to prove generalization bounds on the best function learnable in $\mathcal{H}_B$ and contrast that with $\mathcal{H}_A$, we require a "smoothing" assumption in $\{\boldsymbol{\omega}_i\}_{i=1}^\infty$ (and correspondingly on $\mathcal{H}$). The essence of this assumption is to ensure that the kernel $\mathcal{H}$ can be spanned without introducing much error by only with a few basis components, and is similar to smoothing kernel assumptions made in other areas as well, *e.g.*, in reinforcement learning.

**Assumption 1** ($\gamma$-Exponential Spectral Decay of $\mathcal{H}$). *For the decomposition of $\mathcal{H}$ as outlined in Equation A.4, we assume that there exist absolute constants $C_1 < 1$ and $C_2 = \mathcal{O}(1)$ and parameter $\gamma$ such that $\lambda_i \leq C_1 \exp(-C_2 \cdot i^\gamma)$ for each $i \geq 1$.*

At a high level, our approach is to bound the *test error* of the *empirical risk minimizer* in $\mathcal{H}_A$, with the optimal risk minimizer in $\mathcal{H}_B$ to demonstrate that learning a generalized basis model does not incur significantly larger error compared to learning the full model. We first make these terms precise. Recall that the empirical risk for any function $g$ is given by $\widehat{\mathcal{L}}_n(g) = \frac{1}{n} \sum_{i=1}^n \ell(g(\boldsymbol{x}_i), y_i)$. We denote $\hat{g}$ as the empirical risk minimizer within $\mathcal{H}_B$, *i.e.*,

$$\widehat{g} = \arg\min_{g \in \mathcal{H}_B} \widehat{\mathcal{L}}_n(g). \tag{A.5}$$

Similarly, the *expected risk* can be given, for any function $g$ as $\mathcal{L} = \mathbb{E}_{(\boldsymbol{x},y) \sim \mathfrak{P}}[\ell(g(\boldsymbol{x}), y)]$. Then we can define the *optimal expected risk minimizer* $g_\star$ in $\mathcal{H}_A$ as,

$$g_\star = \arg\min_{g \in \mathcal{H}_A} \mathcal{L}(g). \tag{A.6}$$

We are now equipped to discuss our generalization bound.

**Theorem 1.** *Let $\ell$ be a 1-Lipschitz loss, $\delta \in (0, 1]$ and Assumption 1 hold with constants $C_1, C_2, \gamma$. Then we have that with probability at least $1 - \delta$ there exist absolute constants $C_1, C_2$ such that,*

$$\mathcal{L}(\hat{g}) - \mathcal{L}(g_\star) \leq 2B_{\boldsymbol{w}} \sqrt{\frac{B}{n}} + \frac{DC_2}{C_1} \exp(-B^\gamma) + 5\sqrt{\frac{\log(4/\delta)}{n}}.$$

*Proof.* We will denote the weights and singular values for $f_\star$ as $\boldsymbol{w}^\star$ and $\lambda_{ij}^\star$, *i.e.*, $g_\star(\boldsymbol{x}) = \sum_{i=1}^D w_i^\star h_i^\star(x_i)$ where $h_i^\star(x) = \sum_{j=1}^\infty \lambda_{ij}^\star \boldsymbol{\omega}_j(x)$. Note that this represnetation exists for some $\lambda_{ij}^\star$ by Mercer's Theorem, as discussed earlier. For any $\mathcal{H}_B, \mathcal{H}_A$, consider the function $\tilde{g} \in \mathcal{H}_B$ that is a truncated version of $g_\star$ up to $b$ bases, *i.e.*, $\tilde{g}(\boldsymbol{x}) = \sum_{i=1}^D w_i^\star \tilde{h}_i(x_i)$ where $\tilde{h}_i(x) = \sum_{j=1}^b \lambda_{ij}^\star \boldsymbol{\omega}_j(x)$. Clearly, $\tilde{g} \in \mathcal{H}_B$. We can then rewrite the L.H.S. in the Theorem as,

$$\mathcal{L}(\hat{g}) - \mathcal{L}(g_\star) = \underbrace{\mathcal{L}(\hat{g}) - \widehat{\mathcal{L}}_n(\hat{g})}_{\textcircled{A}} + \underbrace{\widehat{\mathcal{L}}_n(\hat{g}) - \widehat{\mathcal{L}}_n(\tilde{g})}_{\leq 0} + \underbrace{\widehat{\mathcal{L}}_n(\tilde{g}) - \mathcal{L}(g)}_{\textcircled{B}}.$$

Note that the middle term $\widehat{\mathcal{L}}_n(\hat{g}) - \widehat{\mathcal{L}}_n(\tilde{g}) \leq 0$ since $\hat{g}$ is the empirical risk minimizer in $\mathcal{H}_B$. Hence, by bounding terms $\textcircled{A}$ and $\textcircled{B}$, the proof will be complete. We can bound $\textcircled{B}$ via Lemma 1. We have that with probability at least $1 - \delta/2$ for any $\delta \in (0, 1]$,

$$\left| \widehat{\mathcal{L}}_n(\tilde{g}) - \mathcal{L}(g_\star) \right| \leq \frac{L \cdot C_1}{C_2} \exp(-B^\gamma) + 2\sqrt{\frac{\log(2/\delta)}{n}}.$$

We bound $\textcircled{A}$ via bounding the Rademacher complexity [63]. Since the loss function is Lipschitz and bounded, with probability at least $1 - \delta/2, \delta \in (0, 1]$, we have that by Theorem 12 and Theorem 8 of Bartlett and Mendelson [4],

$$\mathcal{L}(\hat{g}) - \widehat{\mathcal{L}}_n(\hat{g}) \leq \mathfrak{R}_n(\ell \odot \mathcal{H}_B) + \sqrt{\frac{8 \log(4/\delta)}{n}}. \tag{A.7}$$

Where $\mathfrak{R}_n$ denotes the empirical Rademacher complexity at $n$ samples [4]. Observe that each element of $\mathcal{H}_B$ is a linear combination of $d$ elements that are represented by $b$ basis vectors in $\mathcal{H}$. Hence, there exist weights $\{\{\alpha_{ij}\}_{i=1}^D\}_{j=1}^B$ such that any $f \in \mathcal{H}_B$ can be written as $\sum_{i,j} \alpha_{ij} \boldsymbol{\omega}_j(x_i), \|\boldsymbol{\alpha}\|_2 \leq B_\mathcal{H} B_{\boldsymbol{w}}$ where $\boldsymbol{\alpha} = \{\{\alpha_{ij}\}_{i=1}^D\}_{j=1}^B$. Furthermore, we have that for any $x$, $\boldsymbol{\phi}(x)^\top \boldsymbol{\phi}(x) = \sum_{j=1}^B \boldsymbol{\omega}_j(x_i)^2 \leq B$. We therefore have, by Theorem 12 of [4] that with probability at least $1 - \delta/2, \delta \in (0, 1]$,

$$\mathcal{L}(\hat{g}) - \widehat{\mathcal{L}}_n(\hat{g}) \leq \mathfrak{R}_n(\ell \odot \mathcal{H}_B) + \sqrt{\frac{8 \log(4/\delta)}{n}}$$

$$\leq 2L\mathfrak{R}_n(\mathcal{H}_B) + \sqrt{\frac{8 \log(4/\delta)}{n}}$$

$$\leq 2LB_{\boldsymbol{w}}\mathfrak{R}_n(\mathcal{H}) + \sqrt{\frac{8 \log(4/\delta)}{n}}$$

$$\leq 2LB_{\boldsymbol{w}}\sqrt{\frac{B}{n}} + \sqrt{\frac{8 \log(4/\delta)}{n}}$$

The last inequality follows from Lemma 22 of [4]. Replacing the above result for $k$, we have that with probability at least $1 - \delta/2$, Using the bound for term $\textcircled{B}$ and applying a union bound provides us the final result.

**Lemma 1.** *The following holds with probability at least $1 - \delta, \delta \in (0, 1]$, for some absolute constant $C \ll 1$,*

$$\left|\widehat{\mathcal{L}}_n(\tilde{g}) - \mathcal{L}(g_\star)\right| \leq \frac{LD \cdot C_1}{C_2} \exp(-B^\gamma) + 2\sqrt{\frac{\log(2/\delta)}{n}}.$$

*Proof.*

$$\widehat{\mathcal{L}}_n(\tilde{g}) - \mathcal{L}(g_\star) = \widehat{\mathcal{L}}_n(\tilde{g}) - \mathcal{L}(\tilde{g}) + \mathcal{L}(\tilde{g}) - \mathcal{L}(g_\star)$$
$$\leq \underbrace{\left|\widehat{\mathcal{L}}_n(\tilde{g}) - \mathcal{L}(\tilde{g})\right|}_{①} + \underbrace{\left|\mathcal{L}(\tilde{g}) - \mathcal{L}(g_\star)\right|}_{②}.$$

To bound ①, we have that for any $(\boldsymbol{x}, y)$ within the training set, $\mathbb{E}[\ell(\tilde{g}(\boldsymbol{x}), y)] = \mathcal{L}(\tilde{g})$ and $0 \leq \ell(\cdot, \cdot) \leq 1$. By Azuma-Hoeffding, we obtain with probability at least $1 - \delta, \delta \in (0, 1]$,

$$\left|\widehat{\mathcal{L}}_n(\tilde{g}) - \mathcal{L}(\tilde{g})\right| \leq 2\sqrt{\frac{\log(2/\delta)}{n}}.$$

For ②, since $\ell$ is $L-$Lipschitz, we have for some $x_1, x_2, y \in \mathcal{Y}$,

$$|\ell(x_1, y) - \ell(x_2, y)| \leq |L \cdot |x_1 - y| - L \cdot |x_2 - y||$$
$$\leq L \cdot |x_1 - x_2|.$$

Therefore:

$$|\mathcal{L}(\tilde{g}) - \mathcal{L}(g_\star)| \leq \left|\mathbb{E}_{(\boldsymbol{x},y)\sim\mathfrak{P}}\left[\ell(\tilde{g}(\boldsymbol{x}), y) - \ell(g_\star(\boldsymbol{x}), y)\right]\right|$$
$$\leq \mathbb{E}_{(\boldsymbol{x},y)\sim\mathfrak{P}}\left[|\ell(\tilde{g}(\boldsymbol{x}), y) - \ell(g_\star(\boldsymbol{x}), y)|\right]$$
$$\leq L \cdot \mathbb{E}_{(\boldsymbol{x},y)\sim\mathfrak{P}}\left[|\tilde{g}(\boldsymbol{x}) - g_\star(\boldsymbol{x})|\right]$$
$$\leq L \cdot \sup_{\boldsymbol{x}\in\mathcal{X}} |\tilde{g}(\boldsymbol{x}) - g_\star(\boldsymbol{x})|.$$

Observe now that for any $\boldsymbol{x} \in \mathcal{X}$,

$$|\tilde{g}(\boldsymbol{x}) - g_\star(\boldsymbol{x})| = \left|\sum_{i=1}^D \sum_{j=1}^\infty (\lambda_{ij}^\star - \widetilde{\lambda}_{ij})\boldsymbol{\omega}_j(x_i)\right| \leq \sum_{i=1}^D \sum_{j=1}^B \left|(\lambda_{ij}^\star - \widetilde{\lambda}_{ij})\right| \leq \sum_{i=1}^D \sum_{j=B+1}^\infty |\lambda_{ij}^\star|.$$

Invoking Assumption 1, we have that

$$\sum_{i=1}^D \sum_{j=B+1}^\infty |\lambda_{ij}^\star| \leq D \sum_{j=B+1}^\infty C_1 \exp(-C_2 j^\gamma) \leq D \int_{j=B}^\infty C_1 \exp(-C_2 j^\gamma).$$

Since $\gamma \geq 1$, we have,

$$\int_{j=r_i}^\infty C_1 \exp(-C_2 j^\gamma) \leq \frac{C_1}{C_2} \exp\left(-B^\gamma\right).$$

A union bound for both parts finishes the proof. $\square$

**Discussion**. The result holds when the target function class is a member of a Reproducing Kernel Hilbert Space (RKHS). While RKHSes include a variety of expressive machine learning function classes, *e.g.*, radial basis functions, polynomials, linear classifiers, it is not known whether arbitrarily initialized neural networks have a small norm in any RKHS with desirable properties. Most notably, however, it was shown recently that certain neural networks can be represented via the Neural Tangent Kernel (NTK), an example of where the theory can be applied as-is. More generally, however, this result demonstrates for arbitrary infinite-dimensional RKHS, we have an exponential dependence on the number of basis $B$ required in the approximation error (second term). Observe that if we set $B = \mathcal{O}(\log D)$, the second term is $o(1)$ and goes to 0 as $n \to \infty$, which suggests that in practice, we only require a number of bases, $B$ that grows logarithmically with the dimensionality $D$.