# OpenReview forum: "Neural Basis Models for Interpretability"
_NeurIPS.cc/2022/Conference — NeurIPS 2022 Accept_

### Official Review · Reviewer_pw3W · 2022-07-10

**Rating:** 5
**Confidence:** 3
**Soundness:** 3 good
**Presentation:** 2 fair
**Contribution:** 2 fair

**Summary:**

This paper proposes a new transparent model called NBM. NBM improves upon NAM, which learns a NN (neural network) for each feature and the final output is a learned weighted-sum of the outputs of the neural networks. Instead of training independent NNs for each feature like NAM, NBM trains a set of NNs as basis and used them across all features.
NBM improves the scalability issue of NAM by reducing the number of independent NNs. NBM is more stable than NAM because of the use of shared NNs.
Experimental results show NBM and its variant NB^2M can get better results than NAM and NA^2M.

**Questions:**

- Do you do any data normalization before training NBM? I didn't remember any related discussions in the paper or supplimentary PDF. If no, when the features' magnitudes vary, does NBM essentially work the same way as NAM since each NN is supposed to work for a particular range of inputs? I believe it's worth to discuss this issue a little bit more.
- Why does NBM achieve better experimental results than NAM? NAM seems to be more powerful in its representative power. Is it the case where NAM achieves better training accurracy/loss but NBM gets better validation/test results? If that's case, does the shared NN work as some implicit regularizations? Will there be some regularizations that make NAM achieve similar results as NBM?

**Ethics Review Area:**

["I don’t know"]

**Limitations:**

- It would be great if the authors can discuss more about what kind of high dimension data can/can't be handled by NBM.

**Strengths And Weaknesses:**

Strengths:
- The proposed model achieved very good experimental results.
- The proposed model is easy to follow.

Weeknesses
- I had to switch back and forth multiple times while reading the paper. For example, the NAM model, an extension of GAM and the predecessor of the proposed NBM, is not mentioned in Section 3.1 Background but in Section 3.3 Dicussion after NBM is introduced. I feel it's more natural to introduce GAM, NAM, then NBM, which helps readers understand the contribution of this paper.
- The discussion regarding the limitation of NBM is quite limited and can be improved. See my questions below.

---

> ### Author Response · Authors · 2022-08-01
> **Author Response to Reviewer pw3W**
>
> - Thank you for your comments, and thank you for stating that “the proposed model achieved very good experimental results”, and that it “is easy to follow”.
> - “It's more natural to introduce GAM, NAM, then NBM, which helps readers understand the contribution of this paper”
>   - Thank you very much for this suggestion, we will revise the manuscript to incorporate it.
> - “Do you do any data normalization before training NBM?”
>   - In Section 4.1 “Datasets” we define the normalization performed before training NBMs. For tabular datasets we perform one-hot encoding for categorical features, and min-max scaling of features to [0,1] range for continuous features. For image datasets, interpretable features are extracted as output probability score (already in [0,1] range) of trained part-attribute models. Same normalization is performed for NAM.
>   - Note that, it is common practice for any approach to normalize structured data [2, 11, 30, 31]. When the normalization is not performed, there is a slight drop in performance for NBM as well as all other baselines. We observe that NBM can handle different ranges of input features by learning particular bases for different inputs.
>   - Finally, in the response to Reviewer 9iGQ we add 9 new datasets, see Table R.1, and follow their instructions for normalization: ordinal encoding for categorical features and quantile transform to Gaussian distribution. With very little hyperparameter tuning, NBMs achieve SOTA results as well.
>   - We will expand on this in our final revision.
> - “Why does NBM achieve better experimental results than NAM?”
>   - Indeed, the Reviewer is right that the representative power of the model class of NAM is larger than NBM, and yes, performance of NBM is better due to it being regularized in the functional space of NAM MLPs. In the Appendix Section A.4, we prove this relationship more precisely in the case when, for any GAM, the shape functions have small norm in a reproducing kernel Hilbert space (RKHS) (Theorem 1 in Appendix). We demonstrate under reasonable regularity assumptions that, the reduction in model capacity, by using a basis model instead of a full GAM, only causes a marginal increase in generalization error at a substantial decrease in model complexity. Moreover, the error term decreases exponentially as the number of bases increases. To the best of our knowledge, this analysis is the first of its kind for interpretable models. This suggests that the basis models can efficiently approximate full GAMs without many basis terms.
>   - Intuitively, the regularization that NBM introduces can be thought of as a functional analog of the regularization that decompositions such as SVD provide. We use only a small set of basis functions to span the space of all MLPs. Indeed, to get NAM to achieve similar performance as NBM, one can do a functional low-rank approximation after the NAM model is trained to regularize it effectively. NBM, however, directly learns this low-rank version during training itself by sharing bases.
>   - Indeed, as Reviewer points out, we often observe that NAM and NBM achieve similar training accuracy / loss, but NBM results in better test accuracy.
> - “What kind of high dimension data can/can't be handled by NBM?”
>   - We do recognize that our approach, though highly scalable, has limitations w.r.t. number of input features. Beyond 10K dense features, or 1M sparse features, we would need to apply some form of feature selection [11, 31, 47] to scale further. Scalability issue is even more pronounced when modeling pairwise interactions in NB$^2$M. However, NBMs can still handle an order of magnitude more than what can be handled by NAM or other GAM approaches that do not perform feature selection. We are exploring a direction where we model higher order interactions via learnable polynomials, which scale significantly better. However, this is beyond the scope of this work, and left for future work.

---

### Official Review · Reviewer_uzMk · 2022-07-11

**Rating:** 5
**Confidence:** 3
**Soundness:** 3 good
**Presentation:** 3 good
**Contribution:** 2 fair

**Summary:**

The paper proposes a Neural Basis Model (NBM) that utilizes basis decomposition of shape function for regression, binary classification and multi-class classification tasks in the context of achieving state-of-the-art accuracy, model size, and throughput. The central idea of the proposed approach is to dissociate the input feature and individually feed it to a set of shared bases that allows for efficient in inference and compact model size.


**Questions:**


1) Have you try the proposed method on the ill-posed tasks? such as image pinpointing and radiance field reconstruction? How does it perform?
2) Is there any similarity between the learned basis and the basis extracted from PCA-like approaches?
3) Any further advantages of interpretable modeling apart from the visualization?

**Limitations:**

The paper lacks a limitations analysis.

**Strengths And Weaknesses:**

What's good:
1) The paper organization and presentation are clear mostly.
2) The idea of decomposing each feature’s shape function into a small set of basis functions seems novel and efficient.

To be improved:
1) Seems the main motivation of the paper is interpretability (as addressed in the title). If so, I would like to see a detailed analysis of the shape of the learned basis and a visualization of how these basis represent the inputs.
2) The comparison approaches are related old I think, how does NBM perform compared with recent implicit functions?
3) The idea of basis learning is also related to the radial basis functions, I would recommend also including some related discussion.

---

> ### Author Response · Authors · 2022-08-01
> **Author Response to Reviewer uzMk**
>
> - Thank you for your comments, and for stating that “idea of decomposing each feature’s shape function into a small set of basis functions seems novel and efficient”.
> - “Analysis of the shape of the learned basis and a visualization of how these basis represent the inputs”
>   - The basis functions are not interpretable themselves, their weighted combination for a given feature is interpretable, as depicted in Figure 3.
>   - We plotted and analyzed the shapes of basis functions, and we observed that the model learns bases of varying frequencies. We will add this analysis in the appendix of the final revision of the paper.
> - “How does NBM perform compared with recent implicit functions”
>   - We added 9 new datasets and compared against the recent state-of-the-art GAM approach NODE-GAM [11]. You can see the comparison and detailed discussion in our response to Reviewer 9iGQ (see Table R.1 and related discussion).
> - “The idea of basis learning is also related to the radial basis functions”
>   - The Reviewer is right that basis learning is related to radial basis functions. Specifically, in the case when shape functions have small norm in a reproducing kernel Hilbert space (RKHS) spanned by an RBF kernel, basis methods correspond to kernel approximation via Mercer’s theorem. We demonstrate in the Appendix A.4 a precise tradeoff (Theorem 1) between the number of basis functions used and generalization error in an RKHS. We will move this discussion to the main paper.
> - “Have you try the proposed method on the ill-posed tasks? such as image pinpointing and radiance field reconstruction?”
>   - We have explored interpretability in image classification and detection (localization) domains. These tasks required significant overhead in proper setup and evaluation in the interpretability area. Thus, we have not explored additional image tasks. However, we appreciate your suggestion, and we will be sure to explore image pinpointing and radiance field reconstruction as future work. If you have suggestions on suitable related works or benchmarks, we would love to add them to our codebase.
> - “Is there any similarity between the learned basis and the basis extracted from PCA-like approaches?”
>   - Yes! The Reviewer is correct - Intuitively, the regularization that NBM introduces can be thought of as a functional analog of the regularization that decompositions such as SVD provide. Instead of learning a basis decomposition of a subspace of $\mathbb R^d$, we optimize via neural networks to select a data-dependent basis of the $L^2$ space of functions ($L^1$ in the case of $\ell_1$-regularized NBM). We will add this point to the final version.
> - “Any further advantages of interpretable modeling apart from the visualization?”
>   - Concept bottleneck models [23], which we demonstrate NBMs improve, are shown to be useful for correction and intervention.
>   - Being able to detect the impact of the bias in the data on the model, and then to repair the model, is critical if we are going to deploy machine learning in applications that affect people’s health, welfare, and social opportunities [5, 10, 44, 45]. This requires models that are interpretable.
> - “The paper lacks a limitations analysis.”
>   - We do recognize that our approach, though highly scalable, has limitations w.r.t. number of input features. Beyond 10K dense features, or 1M sparse features, we would need to apply some form of feature selection [11, 31, 47] to scale further. Scalability issue is even more pronounced when modeling pairwise interactions in NB$^2$M. However, NBMs can still handle an order of magnitude more than what can be handled by NAM or other GAM approaches that do not perform feature selection. We are exploring a direction where we model higher order interactions via learnable polynomials, which scale significantly better. However, this is beyond the scope of this work, and left for future work.
>   - As stated in the conclusions, for the computer vision domain, we assume an intermediate, interpretable concept layer on which GAMs can be applied. We cannot apply NBMs directly on pixel space, while maintaining model interpretability, but we are working on projects that resolve this limitation.
>   - We will add this discussion in the final version of the paper.

---

> > ### Comment · Reviewer_uzMk · 2022-08-09
> > **Final rating**
> >
> > Thanks for the detailed response, I would keep my original rating after reading other reviewers' reviews.
> > Please clearly expose the new table and experiments during revision.
> > BTW, there is a typo in my reviews, it should be inpainting instead of "pinpointing", sorry about the misleading.

---

> > > ### Author Response · Authors · 2022-08-09
> > > **Thank you**
> > >
> > > Thank you for your time and input.
> > >
> > > We will make sure to include following in the camera ready: results on all 16 datasets (9 new from rebuttal) with appropriate baselines and updated SOTA, additional visualizations (some are already added in the updated supplementary, see Appendix Section A.4 and Figure A.2), and additional discussion in response to all reviews.

---

### Official Review · Reviewer_9iGQ · 2022-07-11

**Rating:** 5
**Confidence:** 5
**Soundness:** 1 poor
**Presentation:** 4 excellent
**Contribution:** 2 fair

**Summary:**

# After Rebuttal

I appreciate the authors including multiple datasets in rebuttal, but the performance improvement is still not as big.
The idea is still somewhat incremental in my idea, but the evaluation seems complete and the speed-up looks good compared to NAM. I believe this paper will have more impact if it releases a good codebase, and also shows more GAM graphs in the image datasets.

# Original Review

This paper proposes to improve the NAM model by learning a shared basis function for each feature function i.e. for each MLP of each feature, the first few layers are the same across features. The proposed method, called NBM, allows a strong reduction of the # of parameters without sacrificing the accuracy. It also has higher speed and scales to higher number of features. It also models the interaction term, called NB2M, by modeling each pairwise interaction term through another MLP. The performance of NBM is slightly better than NAM in several tabular and image datasets although consistently.

**Questions:**

1. **Dataset picking**: Why are there several datasets in NAM not compared? Also, image datasets are a very poor use of GAM - no one would want to visualize the GAM plot of a single pixel. I get that the shared basis functions of NBM should work better with images since each feature is homogenous, but again no one would use GAM on images. If the idea is to test NB2M in high-dimensional features, there are multiple high-dimensional tabular datasets used in the NodeGAM paper that can be compared to. Also, why do you mostly include multi-class datasets while NAM uses binary ones?

2. Doesn't the NB2M still have the quadratic growth of the parameters? In Eq. (5), the parameter $b_{i,j,k}$ still grows quadratic.

3. IMHO, the feature selection method in NodeGAM by using attention and back-propogation is not complex, just a type of regular deep learning architecture.

4. It would be nice to show if NBM is better than NodeGAM since it's also a neural GAM and should have fewer #params and faster throughput than NAM already. It also supports multi-class classification unlike EB2M. But I understand it can be perpendicular to the paper's main point to improve from NAM.


[1] Chang, Chun-Hao, Rich Caruana, and Anna Goldenberg. "Node-gam: Neural generalized additive model for interpretable deep learning." arXiv preprint arXiv:2106.01613 (2021).

**Limitations:**

One limitation I can think of is that the hard sharing could lead to negative transfer among feature networks and deteriorates the performance.

**Strengths And Weaknesses:**

# Strengths
- Clear writing.

# Weaknesses
- The datasets, IMO, are poorly chosen in this paper. Several image datasets are used, but no one would use GAM on pixels to claim any interpretability. Also, several original datasets in NAM are not compared. In contrast, some rare datasets like CoverType are used. It makes me wonder if there are dataset picking and the reported performance is not representative.

- Low originality: I believe the idea is not new as sharing basis functions is common, e.g. CNN v.s. MLP and multi-task learning. The basis sharing might work better but the hard sharing in NBM can incur negative transfer that lowers the performance, especially in tabular datasets when the features are very heterogenous. Therefore, I think it's more important the author shows more comprehensive evidences.

- Even with the selected datasets, the performance benefit does not seems to be big.

---

> ### Author Response · Authors · 2022-08-01
> **Author Response to Reviewer 9iGQ**
>
> We would like to thank the reviewer for their remarks. Please note that to address your concern of “poor dataset picking”, we have evaluated NBM against NAM [2] and NODE-GAM [11] on **9 additional datasets** (the 6 largest from the NODE-GAM paper, and 3 missing from the NAM paper), as summarized in Table R.1. NODE-GAM results are reproduced from [11], we use train/val/test from NODE-GAM [11], and perform the same feature normalization: ordinal encoding for categorical features and quantile transform to Gaussian. Even with minimal hyperparameter tuning and no additional model selection (architecture tuning, stochastic weight averaging, etc.) NBMs outperform NODE-GAM on 7 out of 9 datasets (albeit marginally), and outperform NAM on all datasets. As the reviewer themselves noticed, the idea of NBM is perpendicular to that of NODE-GAM and it is possible that bigger gains can be achieved by combining them. We will also revise the paper to include this evaluation. We hope this addresses your concern of dataset selection. Please find more details on this evaluation and answers to all remaining comments below.
>
> ### Table R.1
> | Method   | MIMIC-II (AUC)      | Credit (AUC)        | COMPAS (AUC)        | Click (ERR)         | Epsilon (ERR)       | Higgs (ERR)         | Microsoft (MSE)     | Yahoo (MSE)         | Year (MSE)       |
> | -------- | ------------------- | ------------------- | ------------------- | ------------------- | ------------------- | ------------------- | ------------------- | ------------------- | ---------------- |
> | NAM      | 0.8539 $\\pm$0.0004 | 0.9766 $\\pm$0.0027 | 0.7368 $\\pm$0.0002 | 0.3447 $\\pm$0.0005 | 0.1079 $\\pm$0.0002 | 0.2972 $\\pm$0.0001 | 0.5824 $\\pm$0.0002 | 0.6093 $\\pm$0.0003 | 85.25 $\\pm$0.01 |
> | NODE-GAM | 0.832 $\\pm$0.011   | 0.981 $\\pm$0.011   | **0.742** $\\pm$0.009   | 0.3342 $\\pm$0.0001 | 0.1040 $\\pm$0.0003 | 0.2970 $\\pm$0.0001 | 0.5821 $\\pm$0.0004 | 0.6101 $\\pm$0.0006 | **85.09** $\\pm$0.01 |
> | NBM      | **0.8549** $\\pm$0.0004 | **0.9829** $\\pm$0.0014 | 0.7396 $\\pm$0.0002 | **0.3316** $\\pm$0.0002 | **0.1038** $\\pm$0.0002 | **0.2969** $\\pm$0.0001 | **0.5817** $\\pm$0.0001 | **0.6084** $\\pm$0.0001 | 85.10 $\\pm$0.01 |
>
> - “Dataset picking”
>   - We would respectfully argue the opposite. We have made an effort to select datasets where the difference in the performance between Linear models and full-complexity models (MLP and XGBoost) is the largest, e.g., datasets for which non-linear shape functions and higher-order interactions play a strong role in the optimal solution, and CoverType is a great example of such a dataset. Moreover, we wanted to push our method to the limit, so we evaluated on a dataset with: (i) many input features - Newsgroups has 150k; (ii) many datapoints - Common Objects has 2.5M; (iii) many classes – iNaturalist Birds has 1.5K. Finally, multi-class problems, especially in CV, have been notoriously underexplored in interpretable ML. Nevertheless, the evaluation on 9 extra datasets above should address any concern.
> - “Why are there several datasets in NAM not compared?”
>   - MIMIC-II requires training in order to download and use the dataset, which was not possible before the deadline. The Credit and COMPAS datasets do not have sufficient complexity, i.e., Linear and Non-Linear models give comparable AUC and hence performance differences are often within standard deviation. Nevertheless, we have since added MIMIC-II, Credit, and COMPAS in Table R.1.
> - “Why do you mostly include multi-class datasets while NAM uses binary ones?”
>   - The NAM paper evaluates regression and binary classification, we evaluate on regression, binary classification, and multi-class classification. We noticed that the multi-class problem is more challenging, hence we see this as a merit of our work, not a drawback. Many binary datasets are added in Table R.1.
> - “Multiple high-dimensional tabular datasets used in the NodeGAM”:
>   - We added the 6 large-scale datasets from NODE-GAM [11] in Table R.1.
> - “No one would want to visualize the GAM plot of a single pixel”:
>   - We would like to point out that the reviewer has misunderstood our image datasets evaluation. As mentioned explicitly in Section 4.1 (under “Image Datasets”), we perform concept-bottleneck-style [23] preprocessing, where each image is represented by interpretable features, eg, a “bird” image can be represented with “striped wings”, “needle-shaped beak”, “long legs”, etc that are predicted from the image using a CNN. Then GAMs are fitted on top of the predicted interpretable features, and shape functions can be visualized accordingly, thus there would be NO plots of GAM on a single pixel. This is an established approach in the literature to create interpretable computer vision models, known as the Concept Bottleneck. We encourage the reviewer to consult the seminal work in [23].
>
> Please find response to your other concerns in the subsequent comment below.

---

> > ### Author Response · Authors · 2022-08-01
> > **Author Response to Reviewer 9iGQ (continued)**
> >
> > - “Low originality”:
> >   - To the best of our knowledge, sharing basis functions is a novel concept in the context of GAMs, or interpretability in general. We propose a novel sub-family of GAMs using shared bases, that can be learned in an arbitrary fashion (splines, boosted trees, etc), and an approach to learn them via DNNs. We also propose an efficient first-of-its-kind extension to sparse datasets, where other GAMs do not scale without any feature selection.
> >   - We extensively evaluate on regression and binary classification, as well as on multi-class tabular and image datasets, which have been underexplored in GAM papers: EBM [30, 31], NAM [2], NODE-GAM [11].
> >   - In addition to the algorithmic and experimental contributions, we provide novel learning-theoretic results on functional basis approximation that are presented in the Appendix A.4. We demonstrate a precise guarantee on the generalization error (Theorem 1) that applies to any basis decomposition of GAMs where the shape functions lie in a reproducing kernel Hilbert space (RKHS). To the best of our knowledge, such an analysis is not present in the existing literature on interpretable machine learning.
> > - “The hard sharing in NBM can incur negative transfer that lowers the performance”
> >   - We haven’t observed NBMs obtaining lower accuracy compared to the most related approach NAM in 7 datasets presented in the paper, and additional 9 added in the rebuttal, see Table R.1.
> >   - In addition, in Appendix A.4, we show that we only require $B = O(\log D)$ for a competitive performance. Moreover, using a shared basis acts as a regularizer that makes training more stable. As is with any modeling inductive bias, one must analyze the data prior to learning a model to understand the best hyperparameter setting.
> >   - If the reviewer has a precise theoretical or experimental setting in mind, it would be great to learn a concrete case of “negative transfer”. Across our experimental benchmark (which now contains 16 datasets across tabular tasks as well as structured computer vision problems via concept bottlenecks), we have yet to see any evidence for negative transfer. Moreover, our theoretical results also suggest that with sufficient bases, “negative transfer” is unlikely to occur.
> > - “The performance benefit does not seem to be big.”
> >   - We would like to respectfully disagree: the performance benefit of using NBMs instead of NAMs (the most related model) is four-fold:
> >     - Up to 5% relative improvement in prediction accuracy
> >     - Up to 50x reduction in parameters
> >     - Up to 7x better throughput
> >     - For large datasets, e.g. Newsgroups and Common Objects (with pairwise interactions), NAM does not scale at all. NBM scales effortlessly.
> > - “Doesn't the NB2M still have the quadratic growth of the parameters? In Eq. (5), the parameter $b_{i,j,k}$ still grows quadratic.”
> >   - The reviewer is indeed correct that the $b_{i,j,k}$ grows quadratically, but that parameter is negligible compared to the size of the basis model, i.e., B << N where B is number of bases, and N is #params in the basis model. Entire model of size M in NA$^2$M grows quadratically with feature dimension, while only our smallest part of the model grows quadratically. Nevertheless, we will correct the statement.
> >   - To further emphasize the curse of dimensionality in NA$^2$M vs NB$^2$M, we perform similar analysis as in Eq (6):
> >     - |NA$^2$M| / |NB$^2$M| = 6402    /   (125640/(D(D-1)) + 101)
> >     - This value is 1.0 for D=5 (it was 1.0 for D=10 in NAM vs NBM), and it is 4.28 for D=10, 56.31 for D=100, 63.39 for D=1000, etc.
> >     - I.e., this ratio is much more pronounced when modeling feature interactions.
> >     - We will add this analysis and respective plot in the final revision.
> > - “The feature selection method in NodeGAM by using attention and back-propogation is not complex”
> >   - Thanks for the remark, we will update that statement in our final revision. Our statement in L76-81 does not target NODE-GAM work directly, but aims at motivating our choice of leaving the feature selection approaches out of scope of our work. We will remove “complex” from  “we do not apply complex feature selection algorithms.”
> >   - We want to emphasize once more, that any of the mentioned feature selection algorithms is complementary and can be applied to NBMs.
> > - “It would be nice to show if NBM is better than NodeGAM”
> >   - We compare against NODE-GAM in Table R.1. NODE-GAM results are reproduced from [11]. Identical train/val/test splits are used and same feature normalization is performed in our codebase, to respect fair comparison. Without any feature selection, NBM is comparable to NODE-GAM, and it outperforms it on 7 out of 9 datasets, admittedly with a small margin.
> >   - We will add these results in our final revision, together with NODE-GAM implemented and evaluated on our codebase. Finally, we will also add NB$^2$M and NODE-GA$^2$M comparison, which was omitted here due to a limited timeline for rebuttal.

---

> > > ### Comment · Reviewer_9iGQ · 2022-08-08
> > > **Thank you; Score increases**
> > >
> > > I appreciate the effort authors bring to the revision especially given the short period of rebuttals. The datasets are much more comprehensive which relieves the dataset picking concern. The performance from NAM and NBM seems most in Click but relatively small in others.
> > >
> > > I think it will be very valuable if you can show more GAM graphs (also requested from below reviewer uzMk) especially on the image datasets to see if they are indeed interpretable qualitatively, or maybe not interpretable in some cases. I believe there is no one applying concepts bottleneck + GAM to image spaces and I would love to see if this approach makes sense.
> > >
> > > Overall, I still think the proposed change is still somewhat incremental, and the performance gain does not seem to be much except for the imaging datasets (iNat Birds and Common Objects) and CLICK.
> > > But I do understand the point that improves from the original NAM which makes it faster and more scalable. And the experiments in image datasets are interesting but I believe it needs more analysis.
> > >
> > > One thing that may be worth noting is that Openreview allows modifying the paper, so it will be great to see if those promised changes are incorporated in the revision. But I understand there are space and time issues so it's not necessary at this point.

---

> > > > ### Author Response · Authors · 2022-08-09
> > > > **Additional visualization on image dataset**
> > > >
> > > > Thank you for the updated review and score! We have updated the supplementary material (see Appendix Section A.4 and Figure A.2) to add some GAM graphs on CUB-200 dataset for different bird species. We will add more visualizations for the camera ready.
> > > >
> > > > Likewise, we will include all the results presented during the rebuttal period (all 16 datasets) in the camera ready, we would like to run all baselines (Linear, MLP, etc.) and ensure we meet the space limit for which we need a bit more time, I hope the reviewer can appreciate our effort.
> > > >
> > > > Finally, additional discussion in response to all reviews will be incorporated in the final revision of the paper, as well.

---

### Official Review · Reviewer_3MrK · 2022-07-12

**Rating:** 7
**Confidence:** 3
**Soundness:** 3 good
**Presentation:** 3 good
**Contribution:** 3 good

**Summary:**

This paper proposes a new GAM-like model, denoted as Neural Basis Model (NBM) to analyze the learned feature in machine learning models.  Comparing with the traditional GAM, NBM learns a set of shape functions for each feature using a single neural network. Comparing with NAM, NBM requires less parameters when the dimensionality is greater than ten. The experiment shows NBM can achieve similar or slightly higher results to NAM on multiple datasets using less parameters. The visualized shape functions show that NBM is more stable than NAM, a lower standard deviation is achieved on 100 runs.

**Questions:**

Generally, this draft is well- written and organized, the only question I have is about the "sparse architecture"(line 120). It would be even better to add one more column about the average sparseness of those tabular datasets to Table 1, which could be another factor behind the acceleration shown in Table 2.

**Ethics Review Area:**

["I don’t know"]

**Limitations:**

yes

**Strengths And Weaknesses:**

The number of parameters of NBM is less than NAM when the dimensionality is high, the trick behind is the convolution-like shared shape function (with a fixed small number of B parameters, Eq.(6)), and this NBM can be extend to the bi-variate situation based on Eq(5). This simple yet effective design outperforms the previous state-of-the-art NAM.

---

> ### Author Response · Authors · 2022-08-01
> **Author Response to Reviewer 3MrK**
>
> - Thank you for your comments, and for recognizing our “simple yet effective design” and that our paper is “well- written and organized”.
> - “Sparse architecture” – We evaluate NBMs on two sparse datasets, Newsgroups with sparsity 99.9% (only around 150 words from a vocabulary of 150k words appears per a given article, on average) and Common Objects with sparsity 97% (only around 76 part-attribute compositions from a vocabulary of 2618 compositions are active for a given object, on average). Thank you for your suggestion, we will add the sparseness values in Table 1.

---

### Meta-Review · Area_Chair_Ez5x · 2022-08-30

**Recommendation:** Accept
**Confidence:** Less certain

**Metareview:**

The paper proposes an approach, Neural Basis Model (NBM), that can be seen as a new subfamily of Generalized Additive Models for interpretability. The proposed model is compared to alternatives, showing competitive performance while being computationally more efficient. The authors successfully addressed questions raised by reviewers. As also noted by the authors, a major limitation of the paper remains to be the requirement of the input features being interpretable. This would limit the applicability and utility of the proposed model, limiting the significance of the contribution.

**Award:**

No

---

### Decision · Program_Chairs · 2022-09-14

Accept